

# Evaluation of Windsond S1H2 performance in Kumasi during the 2016 DACCIWA field campaign

Geoffrey E.Q. Bessardon[1], Kwabena Fosu-Amankwah [2], Anders Petersson[3], Barbara J. Brooks[4]

[1] School of Earth and Environment, University of Leeds, Leeds, LS2 9JT, UK
[2] Department of Physics, Kwame Nkrumah University of Science and Technology, Kumasi, Ghana
[3] Sparv Embedded AB, Linköping, Sweden
[4] National Center for Atmospheric Science, School of Earth and Environment, University of Leeds, Leeds, LS2 9JT, UK

*Correspondence to*: Geoffrey E.Q. Bessardon (eegb@leeds.ac.uk)

**Abstract.** Sparv Embedded, Sweden (http://windsond.com) has answered the call for less expensive but accurate reusable
radiosondes by producing the Windsond S1H2. To evaluate the performance of the S1H2, in-flight comparisons between the
Vaisala RS41-SG and Windsond S1H2 were performed during the Dynamics-Aerosol-Chemistry-Cloud Interactions in West
Africa (DACCIWA) project (FP7/2007-2013) ground campaign at the Kumasi Agromet supersite (6°40'45.76''N,
1°33'36.50''W) inside the Kwame Nkrumah University of Science and Technology (KNUST), Ghana campus. The results
suggest a good correlation between the RS41-SG and S1H2 data, the main difference lying in the GPS signal processing and
the humidity response time at a cloud top. Reproducibility tests show that there is no major performance degradation arising
from S1H2 sonde re-use.

## 1 Introduction

Accurate in-situ measurements of tropospheric temperature, pressure, water vapour and wind profiles provide
critical input for numerical weather forecasting and climate models, in the quantification of atmospheric thermodynamic
stability, for the development and application of remote-sensing retrievals, and as an important constraint for atmospheric
process studies. Since the 1930s such measurements have been made by small instrument packages attached to balloons
(Jensen et al., 2016) known as radiosondes; the vertical resolution of the profile being determined by the ascent rate of the
balloon (Martin et al., 2011). The many changes in instrumentation, sounding practices and data processing are discussed at
length by many authors including Haimberger 2007; Vömel et al., 2007; Haimberger et al., 2008; Rowe et al., 2008;
Sherwood et al., 2008; McCarthy et al., 2009; Miloshevich et al., 2009; Seidel et al., 2009; Dai et al., 2011; Hurst et al.,
2011; Thorne et al., 2011; Moradi et al., 2013; Wang et al., 2013; Dirksen et al., 2014; Yu et al., 2015; Bodeker et al., 2016;
Jensen et al., 2016.





The operational cost of launching a radiosonde is high: according to B. Blackmore 2012, personal communication, as cited by Gonzalez et al., 2012, the National Weather Service (NWS) Weather Forecasting Offices (WFO) estimates that the cost per unit launch of a radiosonde in the US is US$ 325 (Price includes radiosonde, balloon and labour) and a total of $21,827,000 a year if 2 launches are made at 92 sites. Martinez (2016) also estimate a cost of US$440,000/year at Summit, Greenland in the remote regions of the Arctic, where only 40 of approximately 1000 routine global launches are made. This would indicate that globally US$ 440M is spent launching radiosondes.

For many years the provision of upper air sounding technology has been dominated by the likes of Vaisala and Graw but over the last decade there has been an increase in the call for less expensive but accurate reusable devices (Douglas, et al., 2012; Martinez 2016; Krauchi and Philipona 2016) and the Windsond S1H2 from Sparv Embedded, Sweden (http://windsond.com) is one such device. This paper compares the performance of this radiosonde with that of established Vaisala RS41.

## 2 The field site

The instrument comparison took place within the framework of the Dynamics-Aerosol-Chemistry-Cloud Interactions in West Africa (DACCIWA) project (FP7/2007-2013) ground campaign at the Kumasi Agromet supersite (6°40'45.76''N, 1°33'36.50''W) inside the Kwame Nkrumah University of Science and Technology (KNUST), Ghana campus: figure 1 shows the location of the field site with respect to the West African Region, Ghana and Kumasi.

The DACCIWA ground campaign has been designed to allow the identification of the controlling processes and factors for low-level clouds LLCs formation and to investigate the low-level clouds (LLCs) effects on the convective boundary layer (CBL). The sounding programme consisted of synoptic sounding at 0600 UTC using a Vaisala (RS41-SG or RS92) radiosonde launched at the Agromet supersite. This time was selected because then the LLC cover was expected to be most intense. In addition to the daily soundings, frequent radiosondes were launched at regular intervals during Intensive Operation Periods (IOPs). The sounding programme had three objectives: 1) to provide the daily statistic of atmospheric conditions 2) to provide more frequent boundary layer radiosounding during DACCIWA IOPs to observe the evolution of the LLCs and associated phenomena such as the Nocturnal Low-Level Jet (NLLJ) and 3) evaluate the Windsond



performance. Figure 2 shows the sounding rationale during DACCIWA IOPs: a single S1H2 launched at 3 AM, two at 6 AM simultaneously with an RS41-SG launch and a final single S1H2 at 9 AM.

The performance comparison between the two systems consisted of: 1) a comparison of the Windsond S1H2 and

Vaisala RS41-SG sondes and 2) an assessment of the reproducibility of the S1H2 during the DACCIWA field campaign.

## 3 The S1H2 Windsond

The Windsond S1H2 is a lightweight (12g) sonde manufactured by Sparv Embedded of Sweden with an operational ceiling of 8 km. Being lightweight the size of the balloon is substantially smaller, a 19-inch "party balloon" being recommended, and hence requires less helium. Like any sounding system, there is a radio receiver. For the Windsond the

RR1-250 Radio Receiver is used and this is connected directly to the host laptop via USB: the arrangement is shown in figure 3. The system has an operational frequency configurable in the range 400 MHz to 480 MHz.

The Windsond launch procedure requires no pre-flight calibration and the firmware in use (v1) allowed up to 4 sondes to be active at any one time. In September 2016, version 2 of the firmware was launched allowing 8 sondes to be active simultaneous while the latest version allows 16.

The operational software provides a "cutdown" feature: when activated the cord attaching the sonde to the balloon is cut. This in conjunction with the integrated instrument retrieval system and prediction of landing site makes the retrieval and reuse of the sonde viable. The S1H2 uses a 1.9g 75mAh rechargeable lithium-ion battery (separate battery): the separated battery allows the sonde to be reused quickly after recovery.

Figure 4 shows Windsond S1H2 and it can be seen that it is based in a styrofome cup: all key features are shown.

Table 1 summarises some of the key physical characteristics of the Windsond S1H2 and the Vaisala RS41, the sonde used for sensor comparison test.

### 3.1 Temperature

Table 2-5 show, on a parameter by parameter scale, a comparison of sensor characteristics. The RS41-SG uses a platinum temperature resistor while a band-gap temperature sensor used in the Windsond S1H2. The silicon band-gap

temperature sensor is a type of thermometer or temperature detector commonly employed in electronic devices. It has good

stability in extreme environmental conditions due to the integral stability of crystalline silicon. Silicon band-gap temperature sensors operate on the principle that the forward voltage of a silicon diode is temperature dependent. Band-gap technology has the advantage of being low cost, accurate and reliable, provide highly consistent measurements, have a positive temperature coefficient with a very low drift over time (Burlet et al. 2015).

Both sensors have the same resolution but the S1H2 has a smaller operational range. The platinum wire temperature sensor of the RS41-SG is both more accurate and has a faster response time than the band-gap sensor (Table 2. Vaisala, 2014 and Windsond Catalogue, 2016).

## 3.2 Humidity

Both sondes use a thin film capacitor to make humidity measurements. These sensors provide a high accuracy,
excellent long-term stability and negligible hysteresis. They are insensitive to contamination by particulate matter, are not permanently damaged by liquids and are resistant to most chemicals. A capacitive humidity sensor works like a plate capacitor. The lower electrode is deposited on a carrier substrate, often a ceramic material. A thin polymer hygroscopic layer acts as the dielectric, and on top of this is the upper plate, which acts as the second electrode but which also allows water vapour to pass through it, into the polymer. The water vapour molecules enter or leave the hygroscopic polymer until the
water vapour content is in equilibrium with the ambient air or gas. The dielectric strength of the polymer is proportional to the water vapour content. In turn, the dielectric strength affects the capacitance, which is measured and processed to give a relative humidity measurement.

The RS41-SG humidity sensor integrates humidity and temperature sensing elements. Pre-flight automatic reconditioning of the humidity sensor effectively removes chemical contaminants in order to improve humidity measurement
accuracy. The integrated temperature sensor is used to compensate the effects of solar radiation in real time. The sensor heating function enables an active de-icing method in freezing conditions during the flight. (Table 3 from Vaisala, 2014 and Windsond Catalogue, 2016).

### 3.3 Pressure

The RS41-SG has a number of variants and particular importance here is the RS41-SG and RS41-SGP. Although both sonde types provide pressure, temperature, humidity and wind measurements it is in the manner in which pressure is derived that the difference arises. The SGP variant has the same pressure sensor as in the RS92 sonde but with revised electronics and calibration while the SG has no pressure sensor at all. In the latter case, the values of atmospheric pressure are calculated from satellite ranging codes, combined with differential corrections from the MW41 ground station. Pressure calculation also uses temperature and humidity from the radiosonde.

The S1H2 measures the pressure with a Microelectromechanical (MEMS) piezoresistor pressure sensor. This technology etches a diaphragm into a silicone substrate. Micro piezoresistors measure the deformation of the diaphragm due to changing pressure.

The difference in performance characteristics (table 4) between the two sondes arise from the S1H2 making direct pressure measurements while those of the RS41-SG are derived indirectly.

### 3.4 Position and winds

Both sondes measure latitude and longitude use onboard GPS receiver pseudorange, however only with the Vaisala system differential correction is applied: the Vaisala ground station being a GPS receiver and the Windsond not. Use of differential GPS techniques in principle improves the accuracy and resolution of measurements.

Both systems determine wind speed and direction independently using the GPS signal without differential correction explaining the two systems similar performance characteristics as seen on table 5.

The Vaisala system determines height using the GPS pseudorange with differential correction while the Windsond uses sonde pressure. The Windsond altitude algorithm tested here does not include hypsometric correction and is corrected in later versions.

## 4 Signal Processing

The Vaisala sounding system MW41 has a single operational mode, unlike the older MW31 which features an operational and a research mode, producing different degrees of signal processing. The MW41 only produces the highest degree of signal processing.

The Windsond S1H2 firmware has a single operational mode and produces uncorrected data. Later versions of Windsond has since introduced data correction of all parameters. During this experiment, ground pressure altitude and temperature have been adjusted to the value measured by the ground-based instrumentation available on the Kumasi supersite.

## 5.  Windsond S1H2 v Vaisala RS41-SG Performance Comparison.

### 130  5.1 Experimental design

### 5.1.1 Profile comparison

The performance of the S1H2 Windsond was assessed by taping S1H2 Windsond and RS41-SG radiosonde together on the same flight at the Kumasi Agromet supersite for the DACCIWA synoptic flight on the 28th of June 2006 launching at 05:44. During this flight, the Windsond S1H2 acquisition was set to three seconds and the Vaisala RS41-SG to

one second. Vaisala RS41-SG data have been reduced to three-second data by selecting measurements taken at the same time as the Windsond S1H2 and only measurements below 6000 m a.g.l have been considered because of the S1H2 recommended operational ceiling. A statistical comparison including linear regression and correlation coefficient between temperature, relative humidity, altitude, wind speed, meridional wind, zonal wind recorded by both sondes was performed. The Windsond S1H2 produces wind speed and wind direction only, the 2-π periodicity of wind direction makes linear

regression irrelevant, so it has been converted to zonal and meridional winds.

### 5.1.2 Signal processing effects in the boundary layer

To analyse the signal processing effect, the same procedure as in 5.1.1 has been performed on the data recorded by the S1H2, the RS41-SG and the RS41-SG after processing from the MW41. The scope has been reduced to data up to 1000





m a.g.l, allowing to see in greater details the difference between the datasets. It also allows direct comparison with the

reproducibility experiment where flights never exceeded 1000 m a.g.l.

### 5.1.3 Pressure comparison

The RS41-SG does not provide raw pressure data so the performance evaluation of the S1H2 pressure sensor is completed by comparing it to the pressure calculated by the MW41 from the RS41-SG data following the procedure described in 5.1.2.

Moreover, the S1H2 altitude measurement uses the pressure sensor data. To assess the influence of the pressure sensor error on the altitude error, the pressure difference between S1H2 pressure and the processed RS41-SG pressure is compared to the difference between the S1H2 and RS41-SG altitude.

During the reproducibility experiment 6, sondes are not attached together and are flying at different ascent rate. To assess the reproducibility of the S1H2, each reproducibility flight data have to be re-aligned to similar vertical level. The

comparison between the pressure and altitude error is used to assess the best vertical level boxes to use in the reproducibility experiment data analysis.

### 5.2 Results

### 5.2.1 Profile comparison

The scatter plot on figure 5 compares respectively temperature, relative humidity, altitude, wind speed, meridional

wind, zonal wind recorded by both sondes, with colours indicating the corresponding altitude according to the RS41-SG. The red line indicates the linear regression between both datasets. For all the assessed meteorological parameter the linear regression parameters are in the range [0.83:1.01] with a correlation coefficient over 0.6 indicating a relatively good agreement between both sondes. However, some discrepancies between parameters or due to sudden atmospheric changes have been identified.

The relative humidity and temperature regression line coefficients on figure 5 (a, b) are within $10^{-2}$ to 1 with correlation coefficient over 0.9, meaning that both sondes are in general agreement over the whole flight. At 2000 m (dark green on figure 5 (a, b)) occurs a sudden temperature increase and relative humidity decrease, and shows discrepancies between sensors. The relative humidity below 2000 m is around 100% indicating the presence of clouds. The sudden heating associated with a sudden drying consequently corresponds to the top of a cloud. For both temperature and relative humidity,

the RS41-SG sensors are detecting the cloud top temperature and humidity changes before the S1H2 sensors. The faster



reply time of the RS41-SG platinum temperature resistor compared to the S1H2 band-gap temperature sensor explains the faster RS41-SG reply to temperature change, while the heating system on the RS41-SG humidity sensor evaporating the cloud water explains the faster RS41-SG reply to relative humidity change.

Wind speed and horizontal wind components, on figure 5 (d, e, f) have the lowest correlation coefficient of all parameters and points are noisy so a smoothing can potentially partially resolve the wind speed and wind component bias. However, the linear regression coefficient below 1 indicates that the S1H2 regularly underestimates the winds. This underestimation can be explained by difference in GPS sensor or the antenna as the Vasaila system does not use differential correction to measure winds.

The correlation between both sensor altitude on figure 5 (c) is the highest of all parameters, while the large root mean square error over 100 and the linear regression coefficient below 1 indicates that the S1H2 regularly underestimate the sonde ascent compared to the RS41. This underestimation can be explained by the absence of hypsometric correction in the S1H2 altitude determination algorithm or/and errors due to the pressure sensor. The influence of the pressure sensor error on altitude error is assessed in section 5.2.3.

### 5.2.2 Signal processing effects in the boundary layer

The scatter plot on figure 6 compares respectively temperature, relative humidity, altitude, wind speed, meridional wind, zonal wind recorded by the S1H2, the RS41-SG and the RS41-SG after processing from the MW41, with colours indicating the corresponding altitude according to the S1H2 with a maximum altitude set to 1000 m. The red line indicates the linear regression between the S1H2 and the RS41-SG data while the blue line indicates the linear regression between the S1H2 and the RS41-SG data after processing from the MW41. A comparison between figure 5 and figure 6, shows that in the boundary layer the correlation between S1H2 and raw RS41-SG is smaller than for the whole profile, this is certainly due to the smaller amount of points considered putting greater emphasis on errors. The comparison of the linear regression coefficient for each parameter on figure 6 shows that the processed RS41-SG data are closer to a 1 for 1 ratio with the S1H2 and the correlation between processed RS41-SG and S1H2 is greater than between the raw RS41-SG and the S1H2. This feature is certainly due to the smoothing operated by the MW41 on the RS41-SG and the adjustment of the maximum relative humidity to 100%. This result shows that the inexpensive Windsond system can reach a level of performance close to the expensive Vaisala system in the boundary layer.

### 5.2.3 Pressure comparison

The scatter plot on figure 7 (a) compares the pressure recorded by the S1H2 and calculated by the MW41 after processing from the RS41, with colours indicating the corresponding altitude according to the S1H2 with a maximum altitude set to 1000 m and the blue line indicates the linear regression between both measured and calculated pressures. The ratio between the pressure measured by the S1H2 and calculated by the MW41 is close to 1 for 1, with an almost perfect correlation and an error below 3 hPa. Comparison of the altitude difference measured by the 2 sondes and the pressure

difference between the calculated and measured pressure shows that over 200 m the pressure difference remains between 2 and 3 hPa while the pressure difference is regularly increasing with height. This shows that the S1H2 pressure sensor error influence on the S1H2 altitude underestimation is small. More recent versions of the Windsond firmware, including hypsometric correction is certainly correcting the altitude bias. The pressure difference consistently remaining between 2 and 3 hPa, thus vertical level boxes of 1hPa are chosen to re-align the sondes during the reproducibility experiment.

**5.3 Windsond S1H2 vs Vaisala RS41-SG Performance comparison conclusions**

The performance comparison between the Windsond S1H2 and the Vaisala RS41-SG shows the potential of the Windsond system which is able to closely match the temperature, pressure and humidity of the Vaisala RS41-SG even after processing by the MW41. However, when a sudden temperature and humidity change happen the slower response time of the Windsond system leads to temporary bias in the profile. The main weakness of the Windsond S1H2 lies into its GPS sensor and antenna which leads to a systematical error in wind speed and components which complicates the observation of phenomenon such as the NLLJ. A more advanced signal processing, can improve the GPS sensor performances. The robust performance of the pressure sensor associated to the altitude systematic error show that corrections in the altitude retrieval algorithm implemented in the latest versions of the Windsond firmware can improve the altitude measurement. The consistent pressure measurements, is leading to use pressure level as the vertical reference to compare the Windsond  S1H2 and the Vaisala RS41-SG during the reproducibility experiment.

**6. S1H2 Windsond Reproducibility Experiment**

**6.1 Experimental design**

The assessment of a sonde reproducibility is essential to guarantee the reliability of the sounding data during the data analysis: alterations of the sonde performance under different atmospheric conditions have to be taken into account for a complete understanding of the data. The re-use feature of the S1H2 requires an evaluation of the data alteration due to the sonde re-use in addition to the reproducibility evaluation using new sondes under different atmospheric conditions.

To complete both assessments, sondes have been launched and retrieved until they got lost. To ensure, according to the authors, the best compromise between ensuring a satisfying recovery rate and a full LLC coverage, the cut-off was set at an altitude of 650 m AGL. At the preset cut-off altitude, two heating coils are activated and the string connecting the sonde to the balloon burnt through. During the sonde descent, until the sonde loses contact with the ground station at approximatively 100 m AGL, the system automatically backs up the trajectory and the predicted landing point in a .kml file.

The ground station was carried to the predicted location, on getting closer, approximately within 50 meters, the

contact between the sonde and the ground station was established and loud beeps (about 15 seconds time interval) and

flashes of light started immediately. Signal strength increased when approaching the sonde and the vice versa. Once

retrieved the sonde was switched off.

        When re-using sonde the cup and lid were checked for any physical damage. The lid of the cup was then opened to

confirm if there are no physical damages to any part (i.e. the heating coils or the printed circuit board PCB). A 4 m polyester

string (sewing thread) was wound around a cardboard (4×2×0.3 cm) cut-out with the ends left free: one to attach to the

balloon the other to tie to the heating coil.

        The sonde renewal strategy has been based on the sonde damage or loss. If a sonde has been lost or any physical

damages were not amendable for the next routine flight a new sonde has been introduced. This strategy has been chosen to

fully evaluate the degradation of the sonde, in terms of both retrieval and data quality but reduced the number of

reproducibility flights with new sondes. The number of times each sonde has been flying as well as the sonde recovery

success are detailed in Table 6. The results will be analysed and associated with the different reasons for a sonde loss.

        Flights, where an S1H2 has been launched simultaneously with another RS41-SG, have been selected for the

reproducibility and data alteration study. During the simultaneous flights, the RS41-SG and S1H2 were attached to different

balloons and consequently not climbing at the exact same ascent rate. The comparison of each pair requires the data to be

aligned at the same vertical level and the systematic underestimation of the altitude by the S1H2 associated to the robust

performances of the S1H2 pressure sensor led to the use of 1 hPa pressure ranges. For each pair, temperature, relative

humidity, total, zonal and meridional winds have been boxed in the pressure ranges. The pairs have been then sorted by the

number of time the S1H2 have been used and the median value for each range and S1H2 number of use have been computed

before a similar statistical comparison is performed on the median values.

**6.2 Results**

        Table 6 details the sonde flight number, the flight success and the sonde recovery for each flight. More than 70% of

the sonde launches have been recovered with the sonde 468 being used 8 times. The recovery rate could have been improved

with more experience using the system and if the receptor had not been damaged due to the difficulties of carrying a laptop





with an antenna in the tropical rainforest and different hazards such as tropical animals. The radio receiver RR2 with

Bluetooth connection seems promising for soundings in a difficult or harsh environment to overcome these difficulties.  Only

5 flights have been identified as unsuccessful showing the overall robustness of the S1H2 radio antenna through the

experiment.

        The scatter plot on figure 8 compares respectively temperature, relative humidity, altitude, wind speed, meridional

wind, zonal wind recorded by the S1H2, and the RS41, boxed in 1 hPa range and sorted according to the number of

soundings of the S1H2 as indicated by the different markers, with colours indicating the corresponding altitude according to

the RS41-SG with a maximum altitude set to 1000 m AGL. The presence of data over 650 m AGL is explained by some

failure of the cut-off system leading to the loss of the sonde but supplementary data for the comparison. For every parameter,

the different markers are superposed randomly indicating the absence of performance degradation over time with the use of

the S1H2 system. However, the sonde S1H2 464 used for the 6th time systematically underestimates relative humidity and

overestimates meridional wind but the sonde 468 used for the 8th time does not show a particular anomaly suggesting a

contamination of the 464 sonde relative humidity sensor. Temperature and relative humidity of sonde 468 during its 8th flight

at 800 m AGL (yellow) show the presence of a cloud top where the lag in the S1H2 answer is identified as in the

performance flight.

Figure 9 shows the linear regression coefficient and the correlation between the boxed S1H2 and the RS41-SG data

for each number of use. For temperature and altitude, the markers are superposed while for the other parameters markers are

more spread but no clear trend can be identified. The sonde 464 used for the 6th time low correlation and linear regression

coefficient for relative humidity and large meridional speed linear regression coefficient confirms the contamination

damaged on the sonde identified in figure 8. The relative humidity low correlation of the sonde 468 used for the 8th can be

explained by the cloud top found in figure 8. The low or negative linear regression coefficient values for speed confirms the

lack of accuracy met in the performance flight and underline a need for improvement in the wind speed calculation from the

GPS data.



### 6.3 S1H2 Windsond Reproducibility experiment conclusions

The reproducibility experiment showed the robustness of the recovery system as well as the sensors. No clear
performance degradation have been identified through the flights and the sondes have been recovered up to 7 times. Similar
performance weaknesses have been identified such as the GPS sensor correction and the sensitivity abrupt temperature and
humidity changes.

However, the maximum altitude has been limited to 650 m AGL to ensure a satisfactory recovery rate which limits
the use of the sonde recovery feature, and a sonde at its 6[th] use showed sign of contamination. A check of the sonde sensors
values with ground instrumentation is consequently necessary before reusing the sonde to increase the confidence in the
measurement.

### 7 Summary and conclusions

The Windsond S1H2 has been developed with the goal of providing an immediate view of local conditions at
different altitudes with a focus on portability and low operating costs to simplify a frequent use in the field. In order to
characterise the performances of the Windsond, an intercomparison flight has been undertaken at the Agronet supersite in
Kumasi, Ghana on the 28th of June 2016. The results show that most of the data recorded below 6000m are in agreement.
However, abrupt changes in temperature and humidity show that the Windsond needs a longer answer time for these
changes. Wind speed and components relatively low performance shows that the GPS sensor and its antenna is a weakness
of the current system.

In the boundary layer, the RS41-SG data processing increase the agreement with the S1H2 data showing that the
expensive Vaisala system performance can be approached by the low-cost S1H2 system. The pressure calculated by the
MW41 from the RS41-SG data are in good agreement with the MEMS pressure senor from the S1H2. The robust
performance of the S1H2 pressure sensor shows that error on the altitude estimation is mainly due to the absence of
hypsometric correction in the retrieval algorithm that current version of the firmware should have corrected. It is therefore
recommended that further performance evaluation of the sonde with a more recent version of the firmware to be conducted.

A reproducibility experiment has been undertaken to assess both the performance of the sonde performance under
different atmospheric conditions and the data degradation due to the sonde re-use. Some of the simultaneous flights were
performed with sondes used several times. The results show that there is no real causality between correlation or ratio

between the sonde changes and re-use of a sonde showing there is a minor degradation in the data accuracy for re-used

sondes. However, one sonde showed contamination signs on the relative humidity sensor. The authors recommend to

compare the sonde performance with ground instrumentation before re-using the sonde.

The capacity of using the same sonde up to 8 times in such a mixed environment as Kumasi constitutes a success

for the Windsond recovery system. However, the author would have wished a louder beep to help recovery in a noisy

environment and also a vibrating system to help the sonde to fall off trees when the sonde, unfortunately, is stuck on it.

The overall success of this experiment shows the potential of this new technology. It is therefore recommended that further

experiments assess quantitatively the reproducibility of the sonde to be conducted in a different environment.

**Author contribution**

Geoffrey E.Q. Bessardon and Kwabena Fosu-Amankwah designed the experiments and carried them out under the

supervision and advice of Barbara J. Brooks. Geoffrey E.Q. Bessardon performed the data analysis. Anders Peterson

provided valuable Windsond system information to perform the analysis. Geoffrey E.Q. Bessardon prepared the manuscript

with contributions from all co-authors.

**Acknowledgements**

The research leading to these results has received funding from the European Union 7th Framework Programme (FP7/2007-

2013) under Grant Agreement no. 603502 (EU project DACCIWA: Dynamics-Aerosol-Chemistry-Cloud Interactions in

West Africa). Both systems used in this research have been provided by NCAS-AMF.

**Competing interests**

The authors declare that they have no conflict of interest.





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

| Sonde Characteristics | RS41-SG radiosondes | S1H2 Windsond |
|---|---|---|
| Weight | 109g | 13 g |
| Dimensions | 272 x 63 x 46 mm | 90 x 75 x 75 mm |
| Battery type | Lithium, nominal 3 V (integrated) | Rechargeable lithium-ion (separate battery) |
| Battery capacity | > 240 min | > 60 min sounding and two days in recovery mode |
| Transmitter power | Min. 60 mW | max 100 mW |
| Telemetry range | 350 km | 60 km |
| Measurement cycle | 1 s | 1 s |

**Table 1 Summary of key physical characteristics of the RS41 and the Windsond S1H2 (based on Table 5 from Vaisala, 2014 and Windsond Catalogue, 2016)**




| Sonde Characteristics | RS41-SG radiosonde | S1H2 Windsond |
|---|---|---|
| **Temperature** | | |
| Sensor type | Platinum resistor | Band gap |
| Measurement range | +60 °C to -90 °C | +80 °C to -40 °C |
| Accuracy repeatability in calibration | 0.1 °C | 0.3 °C |
| Resolution | 0.01 °C | 0.01 °C |
| Response time (63.2%, 6 m/s flow, 1000 hPa) | 0.5 s | 5 s |

**Table 2 Sondes temperature sensor manufacturer specifications (based on Table 1 from Vaisala, 2014 and Windsond Catalogue, 2016)**




| Sonde Characteristics | RS41-SG radiosondes | S1H2 Windsond |
|---|---|---|
| **Humidity** | | |
| Sensor type | Thin-film capacitor, integrated T sensor and heating functionality | Capacitive |
| Measurement Range | 0-100% RH | 0-100% RH |
| Accuracy repeatability in calibration | 2.0% RH | 2.0 % RH |
| Resolution | 0.1 % RH | 0.05 % RH |
| Combined uncertainty in sounding | 4% RH | Not Available (to be assessed) |
| Reproducibility in sounding | 2% RH | Not Available (to be assessed) |

**Table 3 Humidity sensor manufacturer specifications (based on Table 2 from Vaisala, 2014 and Windsond Catalogue, 2016)**






| Sonde Characteristics | RS41-SG radiosondes | S1H2 Windsond |
|---|---|---|
| **Pressure** | | |
| Sensor type | GPS-derived | MEMS pressure sensor |
| Range | Surface to 3hPa | 1100 - 300 hPa |
| Accuracy | Defined as combined uncertainty and reproducibility | 1.0 hPa |
| Resolution | 0.01 hPa | 0.02 hPa |
| Combined uncertainty in sounding | 1.0>100 hPa<br>0.3<100 hPa<br>0.04<10 hPa | Not Available (to be assessed) |
| Reproducibility in sounding | 0.5>100 hPa<br>0.2<100 hPa<br>0.04<10 hPa | Not Available (to be assessed) |

**Table 4 Pressure sensor manufacturer specifications (based on Table 3 from Vaisala, 2014 and Windsond Catalogue, 2016)**




| Sonde Characteristics | RS41-SG radiosondes | S1H2 Windsond |
|---|---|---|
| **Wind** | | |
| Wind speed range | 0-160 m/s | 0-150 m/s |
| Wind speed accuracy | 0.15 m/s | ca 5% |
| Wind speed resolution | 0.1 m/s | 0.1 m/s |
| Wind direction range | 0-360 degree | 0-360 degree |
| Wind direction accuracy | 2 degrees | Depends on GPS conditions |
| Wind direction resolution | 0.1 degree | 0.1 degree |
| Wind velocity uncertainty | 0.15 m/s | Not Available (to be assessed) |
| Wind direction uncertainty | 2 degree | Not Available (to be assessed) |

**Table 5  Sondes wind measurement characteristics (based on Table 7 from Vaisala, 2014 and Windsond Catalogue, 2016)**

| Date and Time | Number of sondes | Sonde A ID | Sonde use number | Successful (Yes/No) | Recovery (Yes/No) | Sonde B ID | Sonde use number | Successful (Yes/No) | Recovery (Yes/No) |
|---|---|---|---|---|---|---|---|---|---|
| 10/06/2016 13:09 | 1T | 471 | 1 | Yes | No | | | | |
| 18/06/2016 03:23 | 1 | 465 | 1 | Yes | Yes | | | | |
| 18/06/2016 05:46 | 2+RS41 | 343 | 1 | Yes | No | 470 | 1 | Yes | No |



| Date/Time | | | | | | | | | |
| --- | --- | --- | --- | --- | --- | --- | --- | --- | --- |
| 21/06/2016 02:43 | 1 | 465 | 2 | No | No | | | | |
| 21/06/2016 08:49 | 2 | 335 | 1 | Yes | No | 466 | 1 | Yes | No |
| 26/06/2016 03:00 | 1 | 468 | 1 | No | Yes | | | | |
| 26/06/2016 05:51 | 2+RS41 | 464 | 1 | Yes | Yes | 376 | 1 | Yes | Yes |
| 26/06/2016 08:36 | 1 | 355 | 1 | Yes | No | | | | |
| 28/06/2016 05:44 | 1R | 305 | 1 | Yes | No | | | | |
| 29/06/2016 02:44 | 1 | 464 | 2 | Yes | Yes | | | | |
| 29/06/2016 05:46 | 2+RS41 | 467 | 1 | Yes | Yes | 468 | 2 | Yes | Yes |
| 29/06/2016 08:43 | 1 | 376 | 2 | Yes | Yes | | | | |
| 01/07/2016 02:37 | 1 | 467 | 2 | Yes | No | | | | |
| 01/07/2016 05:21 | 2+RS41 | 468 | 3 | Yes | Yes | 376 | 3 | Yes | Yes |





| | | | | | | | | | |
|---|---|---|---|---|---|---|---|---|---|
| 01/07/2016 08:35 | 1 | 464 | 3 | Yes | Yes | | | | |
| 03/07/2016 02:49 | 1 | 376 | 4 | Yes | No | | | | |
| 03/07/2016 05:41 | 2+RS41 | 468 | 4 | Yes | Yes | 472 | 1 | Yes | Yes |
| 03/07/2016 08:44 | 1 | 464 | 4 | Yes | Yes | | | | |
| 08/07/2016 02:39 | 1 | 464 | 5 | Yes | Yes | | | | |
| 08/07/2016 05:44 | 2+RS41 | 468 | 5 | Yes | Yes | 472 | 2 | Yes | Yes |
| 08/07/2016 08:40 | 1 | 348 | 1 | Yes | No | | | | |
| 11/07/2016 02:48 | 1 | 468 | 6 | Yes | Yes | | | | |
| 11/07/2016 05:46 | 2+RS41 | 472 | 3 | Yes | Yes | 464 | 6 | Yes | No |
| 11/07/2016 08:40 | 1 | 411 | 1 | Yes | Yes | | | | |
| 14/07/2016 02:59 | 1 | 468 | 7 | Yes | Yes | | | | |





| | | | | | | | | | |
|---|---|---|---|---|---|---|---|---|---|
| 14/07/2016 05:46 | 2+RS41 | 411 | 2 | Yes | Yes | 472 | 4 | Yes | Yes |
| 14/07/2016 08:47 | 1 | 374 | 1 | Yes | Yes | | | | |
| 18/07/2016 05:54 | 2+RS41 | 468 | 8 | Yes | No | 472 | 5 | Yes | Yes |
| 18/07/2016 08:50 | 2 | 411 | 3 | No | No | 374 | 2 | Yes | Yes |
| 21/07/2016 05:55 | 2+RS41 | 382 | 1 | Yes | Yes | 346 | 1 | Yes | Yes |
| 21/07/2016 09:13 | 2 | 472 | 6 | Yes | Yes | 374 | 3 | Yes | Yes |
| 24/07/2016 05:48 | 2+RS41 | 382 | 2 | No | Yes | 346 | 2 | Yes | Yes |
| 24/07/2016 09:23 | 2 | 469 | 1 | Yes | No | 386 | 1 | Yes | Yes |

**Table 6 Table listing sounding time, the corresponding number of radiosonde S1H2 launched. (T denotes the test sonde, R denotes the S1H2 launched taped to an RS41-SG, +RS41 denotes simultaneous launched with the Kumasi Agromet supersite), the sonde id, the number the number of time the sonde has been used, the flight result and the recovery result.**





**Figure 1 Location of the field site with respect to Africa, the West African Region, Ghana and Kumasi**





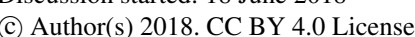


**Figure 2  Scheme representing the sonde routine strategy during DACCIWA IOPs, with RS41-SG (blue) and Windsonde S1H2-R (red)**

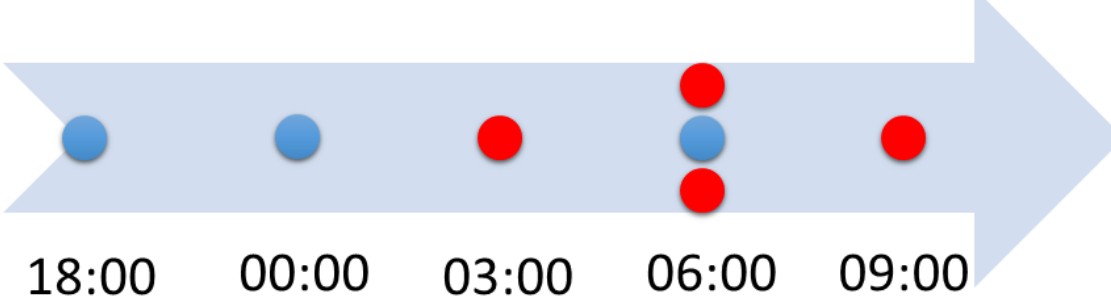

**Figure 3 Experimental system setup: antennae, sounding system, and ground check system (MW41)**




**Figure 4 External shot of the S1H2**





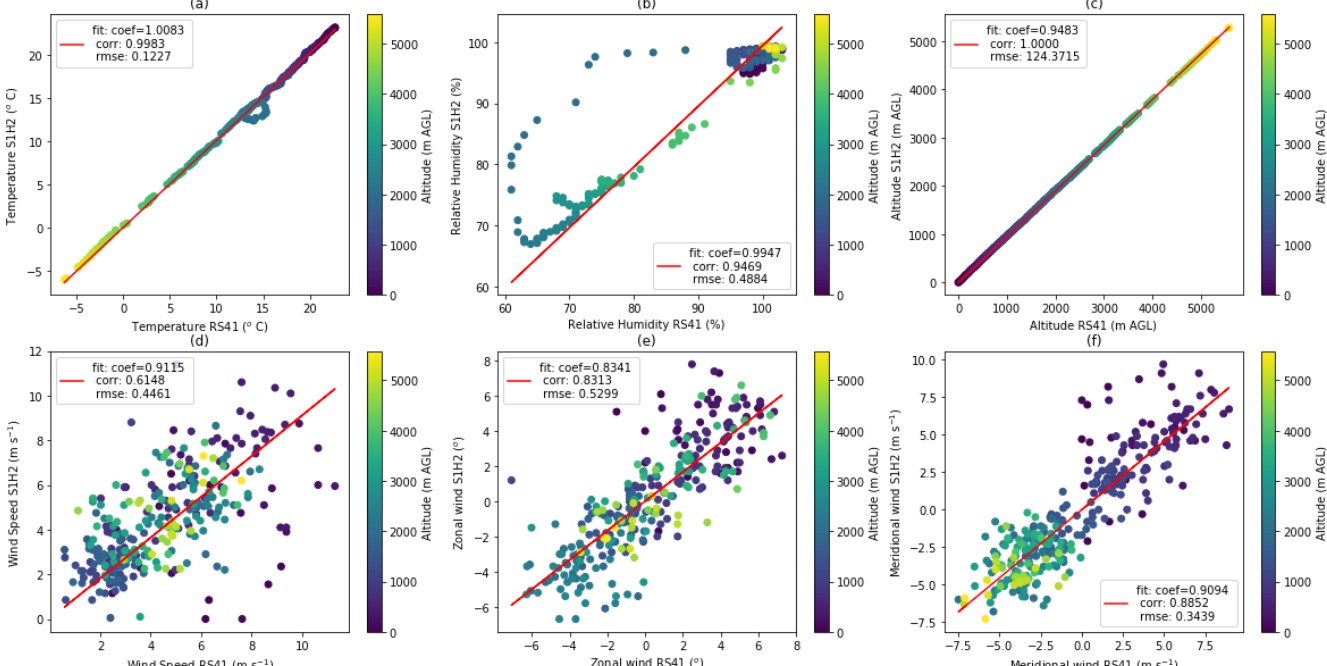

**Figure 5 Comparison of temperature (a), relative humidity (b), altitude (c), wind speed (d), zonal winds (e) and meridional winds (f) recorded by the Windsond S1H2 and the Vasaila RS41-SG during the flight of the 28/06/2016 05:44 in Kumasi. The colors are based on the Vaisaila RS41-SG measured altitude with the maximum altitude set to 6000 m. The red lines indicate the linear regression of each parameter.**





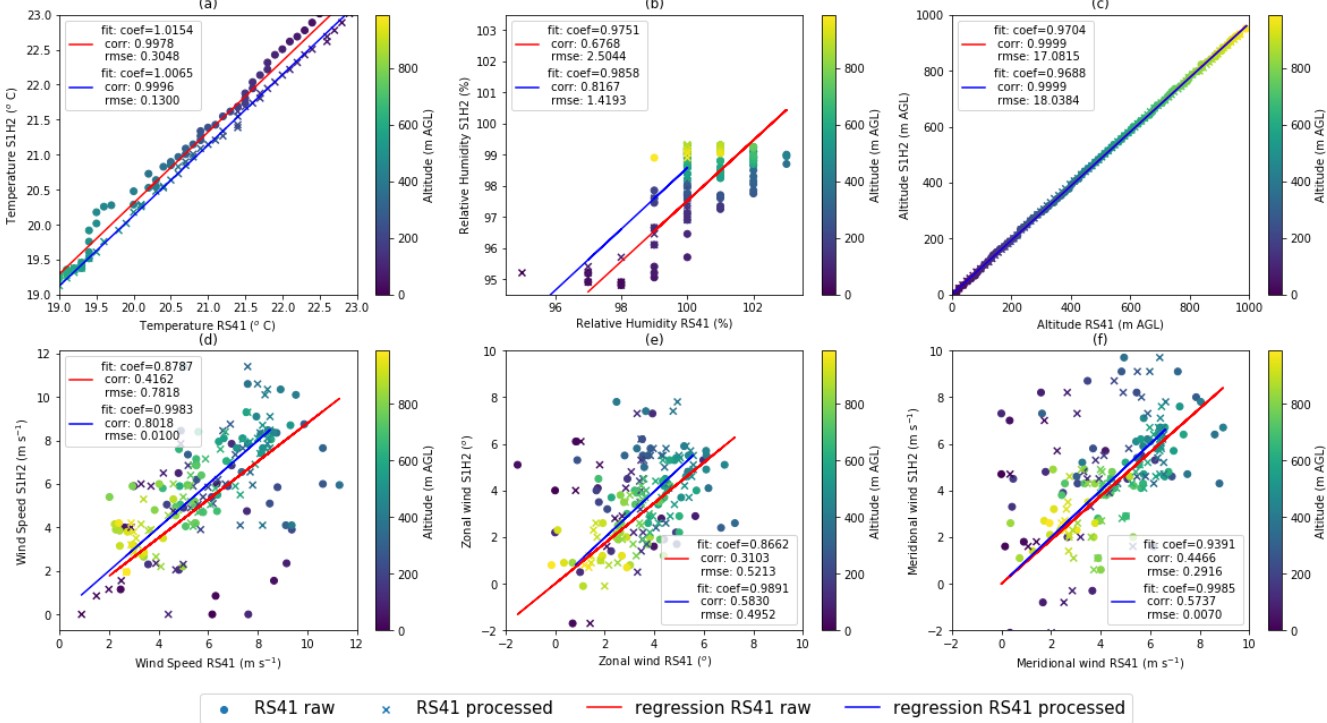


**Figure 6 Comparison of temperature (a), relative humidity (b), altitude (c), wind speed (d), zonal winds (e) and meridional winds (f) recorded by the Windsond S1H2 and the Vasaila RS41-SG before and after processing during the flight of the 28/06/2016 05:44 in Kumasi. The colors are based on the Vaisaila RS41-SG measured altitude with the maximum altitude set to 1000 m.**






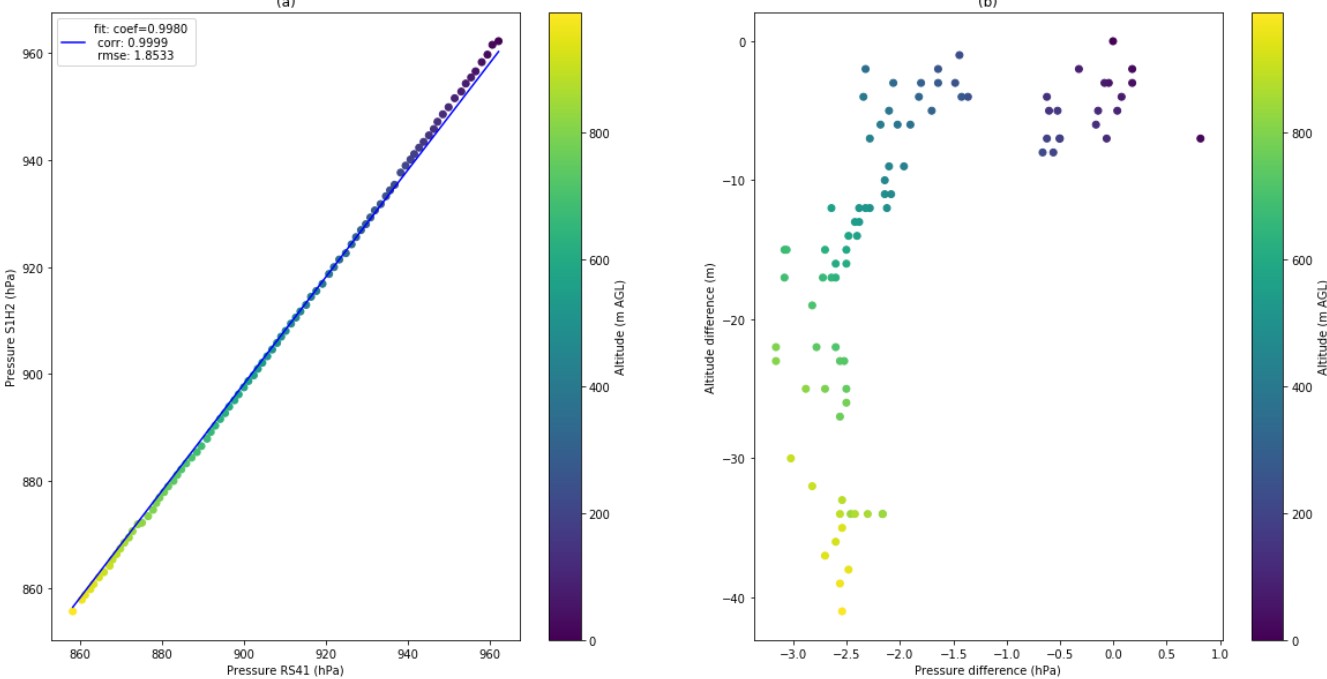

**Figure 7 Comparison of pressure recorded by the Windsond S1H2 and calculated by the Vasaila MW41 (a), the pressure difference between the recorded Windsond S1H2 and the Vaisala MW41 and the altitude difference between the Windsond S1H2 and the Vaisaila RS41-SG (b) during the flight of the 28/06/2016 05:44 in Kumasi. The colors are based on the Vaisaila RS41-SG measured altitude with the maximum altitude set to 1000 m.**






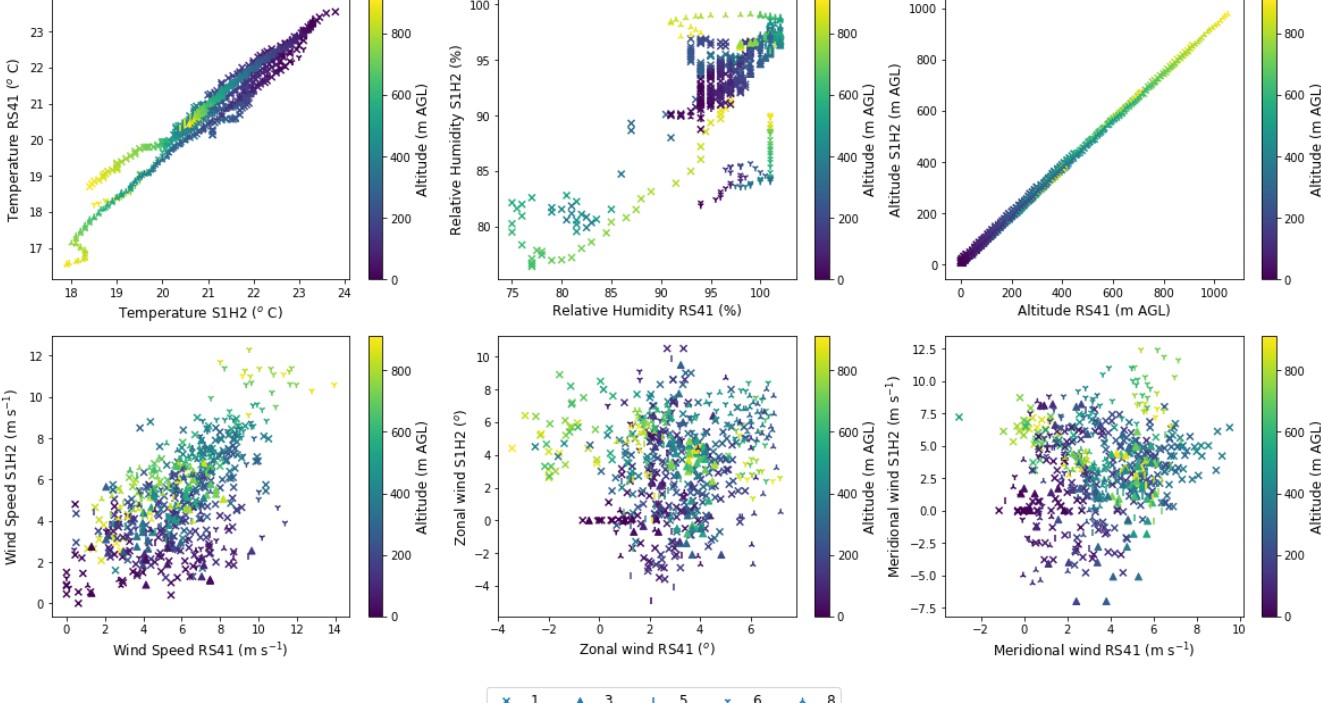

**Figure 8 Comparison of temperature (a), relative humidity (b), altitude (c), wind speed (d), zonal winds (e) and meridional winds**
**(f) recorded by the Windsond S1H2 and the Vasaila during the DACCIWA field camaign in Kumasi. Each marker corresponds**
**to the median value over 1hPa range for all the flights where the S1H2 was used respectively for the 1st, 2nd, 3rd, 4th, 5th, 6th and 8th**
**time. The colors are based on the Vaisaila RS41-SG measured altitude with the maximum altitude set to 1000 m.**



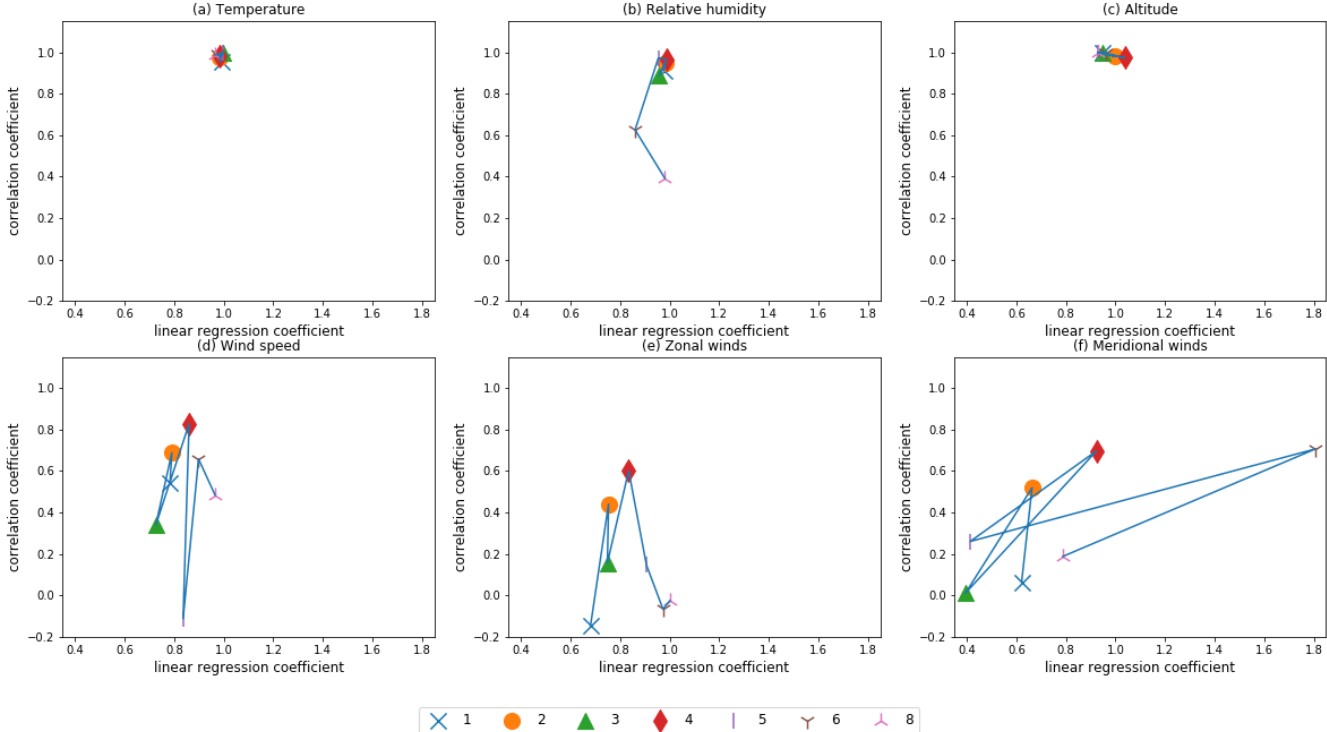

**Figure 9 : Comparison of the correlation coefficient and the linear regression coefficients between the S1H2 and the RS41-SG temperature (a), relative humidity (b), altitude (c), wind speed (d), zonal winds (e) and meridional winds (f) for all the flights where the S1H2 was used respectively for the 1st, 2nd, 3rd, 4th, 5th, 6th and 8th time.**