# Peer review of "Evaluation of Windsond S1H2 performance in Kumasi during the 2016 DACCIWA field campaign"

_Atmospheric Measurement Techniques, 2018_

## Referee Comment (RC1) · Anonymous Referee #1 · 19 Jul 2018

I have several concerns about the paper which I will address in the full review process.

---

## Referee Comment (RC2) · Anonymous Referee #1 · 2 Aug 2018

**Full Review for AMT-2018-179 (Bessardon et al.)**

**Title:** Evaluation of Windsond S1H2 performance in Kumasi during the 2016 DACCIWA field campaign

**Authors:** Geoffrey E.Q. Bessardon, Kwabena Fosu-Amankwah, Anders Petersson, Barbara J. Brooks

**Overall comments:**

This paper describes progress towards developing a less expensive but reliable upper-air radiosonde. To evaluate their newly developed S1H2 sonde they compare its data to observations from high-quality Vaisala RS41-SG sondes. The observations were taken from 33 launches during the DACCIWA field campaign in Western Africa. Basically the authors conclude that the S1H2 sonde is a work in progress with the main issues being the poor performance of the GPS sensor leading to questionable winds and the slow response time of the temperature and humidity sensors. It's ironic that an instrument called a "windsond" would do such a poor job measuring winds. They conclude by offering some recommendations for future improvements.

From the limited comparisons shown between RS41 and S1H2 observations, it is hard to properly judge the performance of the windsond. For example, only one intercomparison flight is made for data extending above the boundary layer. Figures 5-7 show data from this one flight. To get meaningful statistics to evaluate the windsond, data from 20 or more flights should be presented as in Jensen et al. (2016) and similar intercomparison studies. For soundings within the boundary layer, analyses are shown from (I believe) eleven flights (Figs. 8-9) and in a format that is difficult to interpret. I would recommend that analyses be presented in a more conventional format as biases and rms differences between the RS41 and windsond (see Fig. 8 of Jensen et al. 2016).

While the paper has some major concerns in the way the analyses are presented, it is still of value in that it is introducing a new instrument with a promising upside that is in the early stages of development. Under major comments below I suggest several areas where paper could be improved.

**Major comments:**

While the windsond system is being marketed as a less expensive replacement to more conventional sondes, no where is the cost of the sonde system (laptop, antenna, etc.) and sondes mentioned in the paper. Please discuss this information.

Line 24: The vertical resolution is also a function of the sampling rate.

Line 28-33: So the US sites are spending ~$237K per site per year. I would assume that the US sites are some of the more costly ones to maintain around the globe so I would guess your $440M is gross overestimate. You might want to state a range like from $237M to $440M. The statement referencing Martinez (2016) is confusing. It reads as if

you saying that Greenland has 40 operational sites? I'm assuming you mean the Arctic has 40 sites. You may want to reword this statement. Also, is Martinez (2016) a valid reference?

With an operational ceiling of 6 km, it does not seem that the windsond system can be used to replace the sondes currently being used at operational sites which record data to 25 km and higher. With this mind what are the practical research applications of the windsond S1H2 as an upper-air system? Because of its limited range it seems best suited for use in boundary layer studies, however boundary layers are often characterizes with sharp gradients in potential temperature and moisture which the S1H2 has difficulty resolving because of its slow response time. Please discuss.

Are there plans to use improved T and RH sensors with a better response time?

Line 56: Why is the operational ceiling at 8 km? Is this the burst altitude of the party balloon used with the sonde or are there some other considerations?

Figure 4: It's difficult to see the ruler in this picture to get an idea of the length of the sonde.

Line 104: Also mention that the RS41-SG pressure calculation uses the hypsometric equation.

Line 123-124: Please clarify what it means "that the MW41 only produces the highest degree of signal processing". In other places you mentioned RS41 data before and after processing.

Line 126: Please clarify what corrections have been introduced. Have these corrections been implemented in the results from this study?

Line 153: What is experiment 6?

Line 167-168: This discrepancy between sensors at 2000 m is difficult to see in the manner that the data is displayed. Could the data be presented as a function of height or pressure to better show this?

Line 176: Please verify that Vaisala does not use GPS differential correction to compute winds as I thought they did. In fact this statement seems to contradict what is said earlier in lines 111-113. Did you mean the S1H2 does not do differential correction to compute winds. It seems really puzzling why the Windsond winds are of such poor quality. For example the IMET sonde system does not use a differential wind correction and its winds compare quite favorable to the RS41 sonde. Can you give some explanation for the poor performance of the Windsond winds? Is some of this error due to the pendulum motion of the sonde swinging below the balloon which is filtered out in the RS41 processing but not filtered out by S1H2 system?

Line 195: One sounding does not provide statistically significant evidence for this statement. See comments above.

Section 5.2.2. So to clarify are you saying that the results shown for the S1H2 have no post processing and no corrections applied? Can you state what processing and corrections the MW41 performs. You mention smoothing in line 194. Is this smoothing of all fields? Is the balloon pendulum motion only taken out in the MW41 processed data?

Figure 7: It appears that the surface or starting pressure used is different between the systems. Why is this?

Line 207: Does the pressure difference between the two systems continue to increase with altitude?

Line 229: What is a .kml file? Does this need to be mentioned?

Line 232: Are these flashes of light coming from the sonde? Please clarify.

Line 235: Have you considered if a 4m string is long enough to prevent balloon effects on the sonde observations? I believe the Vaisala system uses a much longer string (20-30m) to prevent any balloon impacts on the sonde data.

Line 244: Please clarify what the "data alteration study" is.

Line 285: This is good suggestion and should be a standard practice for all flights (i.e., proper surface base-lining of sondes).

Table 3: Please mention the RH sensor response time.

Listed below are some additional minor suggested changes the authors may want to consider.

**Minor comments**

Line 48: suggested rewording, "because the LLC cover …"

Line 50: suggested rewording, "boundary layer sounding during …"

Line 69: "Figure 4 shows the Windsond …"

Line 74 "sensor is used in …"

Line 134 and elsewhere like Table 6: mention if time is GMT or LT.

Line 160: "all the assessed meteorological parameters …"
Line 168: "sudden warming …"

Line 171 and 172: change "reply" to "response"

Line 234: "When re-using the sonde …"

Line 256: "for locating soundings …"

Line 289: Seems like "different altitudes" should be "lower altitudes". This would be a good place to state the specific niche that the Windsond is trying to fill. Certainly in its current configuration it will never be used as an operational sounding.

Line 292: "longer response time …"

---

## Author Comment (AC1) · 1 Oct 2018

We thank the anonymous referee for the helpful comments. We are responding to all the comments of the reviewers in the supplement and we have prepared a revised manuscript where major changes are marked in red.

Please also note the supplement to this comment: https://www.atmos-meas-tech-discuss.net/amt-2018-179/amt-2018-179-AC1-supplement.zip

---

## Referee Comment (RC3) · Anonymous Referee #1 · 6 Oct 2018

Title: Evaluation of Windsond S1H2 performance in Kumasi during the 2016 DACCIWA field campaign

Authors: Geoffrey E.Q. Bessardon, Kwabena Fosu-Amankwah, Anders Petersson, Barbara J. Brooks

Overall comments:

The authors have offered adequate responses either by addressing my comments or revisions to the paper. However it should be clearly stated in the abstract that this Windsond is intended primarily for collecting boundary layer observations. Also note that boundary layers are typically 500 m over the tropical oceans but can be 5 km deep

under summertime continental conditions. So in the first sentence of the conclusions where you state that it measures conditions at lower altitudes, lists an approximate height range where observations are considered good. For example,"... lower altitudes (up to  $\sim$ 2 km)" or whatever altitude you trust your data.

Finally, in your response you mention that you thought the balloon did not effect the winds. But there is also a concern during daylight flights that radiative effects off the balloon with a short 4m string could effect the T and RH measured by the sonde.

Suggested rewording:

Line 30: This rough estimate varies regionally as the price of labor, helium and balloons is not the same around the globe. Yet operational costs are a significant investment in countries with limited resources.

Line 111-115: "... the Vaisala ground station has a GPS receiver ... However, wind speed and direction are determined independently from the GPS position using the GPS doppler frequency shifts.

Line 117: "Similar to the RS41-SG ..."

Line 206: "... performed. To be statically significant this result needs to be verified with additional performance ..."

Line 239: "During the descent after the sonde loses contact ..."

AMTD

---

## Referee Comment (RC4) · Anonymous Referee #2 · 9 Nov 2018

This paper presents an evaluation of a relatively new low-cost radiosonde system against a well-established and widely used radiosonde based on measurements performed in June and July 2016 during a field campaign in Ghana - Western Africa. The low-cost radiosondes were recovered by the operators and reused up to 8 times, which allows the authors to analyse a relatively high number of ascends. It is shown that under "simple" atmospheric conditions temperature, humidity and pressure measured by both systems compare reasonably well, but as soon as larger vertical inhomogeneities occur the low-cost radiosonde suffers from slow sensor response and hysteresis. GPS-derived wind from the low-cost system is of very bad quality.

Unfortunately, the paper suffers from several weaknesses starting by the design of the measurements, missing technical information, lack of measurements under laboratory conditions and a very limited analysis of the data. The authors miss to cite and discuss relevant literature e.g . Legain et al. 2013 doi:10.5194/amtd-6-3339-2013 and Nash et al, 2010 WMO Report No. 107 Instruments and Observations. The weather situation is not sufficiently discussed and taken into account. Overall it seems to me that the paper is a kind of side product produced with minimal effort.

I think that the paper will not warrant publication as long as a mayor revision is done which addresses the following comments.

Specific comments:

Page 2 The first section is a marketing analysis which is mostly irrelevant if you want to discuss a reusable low cost sonde that is limited to 6000 m altitude. Sonde costs are fixed - price differences for launches in different regions depend on logistics and local labour.

If the sounding program had the objective to evaluate the Windsond performance already from the beginning please explain the following: 1) Why is there only one tandem flight reaching higher altitudes performed 2) Why are all low altitude intercomparison flights performed only at 0600 and not distributed over day and night or at least over the launch times shown on figure 2. As the sondes were recovered no significant additional costs would have been created. 3) Why are the RS42 and Windsond not tied together for the low altitude intercomparison flights – the resulting spacial difference makes it impossible to separate instrument errors from atmospheric variability.

Page 3 Please us UTC or LT but not AM / PM Is Fig. 2 really needed ? Please explain what you mean with simultaneous launched (see above)

Please give more information about the calibration of the Windsond. Do sondes have individual factory calibrations stored on the sonde or does the manufacturer rely on the

[Figure]

quality of its sensors only ? How is the multi sonde reception realized – please give details on the receiver technology. Please use Kelvin instead of °C for Accuracy and Resolution in Table 2

Table 3-5 Anders Petersson is affiliated to the manufacturer of Windsond. You should be able to give detailed information about the sensors used in Windsond and their performance instead of "not available (to be assessed)". Is the given value for pressure accuracy valid for Vaisala or Windsond? Why is the Wind speed accuracy relative to the wind speed?

Page 5: Pressure sections: Please include in the discussion the results of the WMO radiosonde intercomparison 2010 about direct pressure measurements vs. derived pressure.

Page 6: Please explain uncorrected data vs data correction for all parameters for the Windsond. What was the procedure to "adjust" ground pressure altitude and temperature? Wow large were these "adjustments" and why was this not done for humidity?

I am still astonished that only one tandem flight to higher altitudes was performed! A larger number of such flights under different weather situations as well as during day and night would have improved the evaluation significantly. The flight was in 2016 and not 2006. Since all flights were performed during night or early morning radiative effects cannot be evaluated. Experimental design needs to be explained in more detail. What was the length of the line connecting the sondes to the balloon? How did you tape the sondes together? Is it excluded that waste heat of one sonde influenced the other? Why did you set the Windsond acquisition to 3 seconds - according to table 1 the measurement cycle is 1s for both sondes. Please give details about the weather situation.

5.1.2 Should be renamed to Signal processing for low altitudes – Boundary layer higght was not detected - I would expect a boundary layer height around 100 m at the launch time of the sonde rising up to 1500 m during the day in this region during the monsoon

period.

Page 7: Can you explain why you have chosen different ascent rates and non-attached sondes for evaluating the reproducibility ? I can't see any sense in this procedure since natural atmospheric variation will be at least in the same range as the instrument error.

Profile comparison – It would be nice to have a profile plot if you do profile comparisons! Instead of showing scatter plots it would make much more sense to plot vertical profiles of PTH as well as wind for both sondes with an additional profile showing the vertical profile of the difference (Vaisala – Windsond) for each parameter together with the accuracy as stated by the manufacturer's datasheet. This would allow a meteorological interpretation. How do you measure cloud top temperature above the cloud top – The RS41-SG sensors are detecting the cloud top temperature and humidity before the S1H2 …...????

Page 8: Change reply time to response time

The atmosphere is characterized by vertical inhomogeneities, inversions and clouds – radiosondes therefore have to have sensors with low response time and neglectable hysteresis – if this is not the case the sonde is simply not suitable as radiosonde – or only for nice weather well mixed cloud free boundary layer.

Page 9: More recent versions of the Windsond firmware certainly correct the altitude bias - have you checked this? Is it possible to reprocess the measurements to verify? To me it is not shown that newer firmware versions correct the altitude bias.

The conclusions are too favourable – Windsond cannot handle inhomogeneities due to the high response time of the sensors, GPS derived wind error is far above the 5% error given by the manufacturer and to my opinion useless. It is not shown that at least the altitude correction in the latest versions of Windsond improve the systematic altitude error. As the WMO intercomparison results and the Vaisala sonde show pressure sensors are not needed any more for radiosondes – the "robust performance" of the

pressure sensor us unfortunately only of minor importance.

Page 10:

The experimental design shows several weaknesses – as already addressed the fact that the sondes were not tied together during the ascends makes it nearly impossible to separate instrument error and atmospheric variance. I would recommend to test each sonde prior nest launch instead of a simple visual inspection.

I would strongly recommend to perform additional measurements with a larger number of sondes under laboratory conditions to determine sensor accuracy and inertia over a wide range of temperature and humidity and to compare the results to the sondes datasheets first.

Reproducibility can also better be tested in a combination of repeated tandem flights and climate chamber measurements – this would allow the separation of sensor degradation and atmospheric influence in real atmospheric conditions.

Table 6 is unreadable – it extends 4 pages – please consider a condensed way of presentation.

Page 11:

Please give the percentage of unsuccessful flights and flights with sondes that did not cut off. Is the number of data from sondes that did not cut off large enough to do a representative evaluation for altitudes between 650 and 1000 m?

It is nearly impossible to separate the different markers in Fig. 8. Maybe separated figures would help.

As you have a large number of flights over several days available I would recommend to do not only a statistical analysis based on scatter plots and regressions but also a more meteorological where you create classes of different weather situations e.g. with and without low level clouds and analyse the behaviour of the sondes along the vertical

profile.

Fig. 9 – why do you use lines to connect the markers?

Page 12: A check of the sonde sensors before reusing it should be the standard procedure – see my comment to page 10.

A system for low altitude rapid soundings using high quality radiosondes was already introduced and tested by Legain et al. 2013. The questions to me is if a low cost and unfortunately low-quality system like the Windsond really makes sense with all the weaknesses we have seen in your evaluation especially - considering the fact that higher quality sondes can also be recovered and reused so that the cost difference between the sondes gets even less important. Please discuss!

Please change longer answer time to response time.

It would be nice to know if newer Windsond firmware really has corrected the problems with GPS derived wind and pressure. I therefore strongly encourage you to perform the further performance evaluations and include their results into a revised version of your manuscript.

---

## Author Comment (AC2) · 7 Dec 2018

**Title:** Evaluation of Windsond S1H2 performance in Kumasi during the 2016 DAC-CIWA field campaign

**Authors:** Geoffrey E.Q. Bessardon, Kwabena Fosu-Amankwah, Anders Petersson, Barbara J. Brooks

We thank the anonymous referee for the helpful comments. We are responding to all the comments of the reviewer in this document and we have prepared a revised manuscript where changes made for the previous revised manuscript are marked and

in blue and changes for this manuscript are marked in red. In the following, comments of the reviewers are given in bold and italic with our responses are given in normal font.

**Overall comments:** *The authors have offered adequate responses either by addressing my comments or revisions to the paper. However it should be clearly stated in the abstract that this Windsond is intended primarily for collecting boundary layer observations.*

Answer: This information has been added on the first sentence of the abstract

*Also note that boundary layers are typically 500 m over the tropical oceans but can be 5 km deep under summertime continental conditions. So in the first sentence of the conclusions where you state that it measures conditions at lower altitudes, lists an approximate height range where observations are considered good. For example,"... lower altitudes (up to 2 km)" or whatever altitude you trust your data.*

Answer: This information has been added to the first line of conclusion

*Finally, in your response you mention that you thought the balloon did not effect the winds. But there is also a concern during daylight flights that radiative effects off the balloon with a short 4m string could effect the T and RH measured by the sonde.*

Answer: That is true, our answer was focussed on the wind speed error as the wind speed error was the largest. The balloon used under a 4 meter rope for the reproducibility experiment was smaller than the balloon used during the performance flight when the Windsond was taped to Vaisala sondes under the 20 meter string. The T and RH errors during the performance flight are a similar magnitude during the reproducibility experiment so the smaller ballon under a 4 m rope does not seem to have a similar impact on data compared to the larger balloon with a 20 m rope. However, we recognize that the use of a longer string with the smaller balloon would be an in-
expensive way to reduce the radiative effects on the data collected by the Windsond system.

**Suggested rewording:**

All the suggested rewording have been applied to the manuscript

*Line 30: This rough estimate varies regionally as the price of labor, helium and balloons is not the same around the globe. Yet operational costs are a significant investment in countries with limited resources.*

*Line 111-115: "... the Vaisala ground station has a GPS receiver ... However, wind speed and direction are determined independently from the GPS position using the GPS doppler frequency shifts.*

*Line 117: "Similar to the RS41-SG ..."*

*Line 206: "... performed. To be statically significant this result needs to be verified with additional performance ..."*

*Line 239: "During the descent after the sonde loses contact ..."*

[Figure]

**Supplement:**

[revised manuscript text omitted]

Cord

Humidity sensor

On/Off switch

Antenna cable

**Figure 4 External shot of the S1H2**

[Figure]

435

**Figure 5 Comparison of temperature (a), relative humidity (b), altitude (c), wind speed (d), zonal winds (e) and meridional winds (f) recorded by the Windsond S1H2 and the Vasaila RS41-SG during the flight of the 28/06/2016 05:44 UTC in Kumasi. The colors are based on the Vaisaila RS41-SG measured altitude with the maximum altitude set to 6000 m. The red lines indicate the linear regression of each parameter.**

440

[Figure]

**Figure 6 Comparison of temperature (a), relative humidity (b), altitude (c), wind speed (d), zonal winds (e) and meridional winds (f) recorded by the Windsond S1H2 and the Vasaila RS41-SG before and after processing during the flight of the 28/06/2016 05:44 UTC in Kumasi. The colors are based on the Vaisaila RS41-SG measured altitude with the maximum altitude set to 1000 m.**

445

[Figure]

**Figure 7 Comparison of pressure recorded by the Windsond S1H2 and calculated by the Vasaila MW41 (a), the pressure difference between the recorded Windsond S1H2 and the Vaisala MW41 and the altitude difference between the Windsond S1H2 and the Vaisaila RS41-SG (b) during the flight of the 28/06/2016 05:44 in Kumasi. The colors are based on the Vaisaila RS41-SG measured altitude with the maximum altitude set to 1000 m.**

[Figure]

455 **Figure 8 Comparison of temperature (a), relative humidity (b), altitude (c), wind speed (d), zonal winds (e) and meridional winds (f) recorded by the Windsond S1H2 and the Vasaila during the DACCIWA field camapign in Kumasi. Each marker corresponds to the median value over 1hPa range for all the flights where the S1H2 was used respectively for the 1st, 2nd, 3rd, 4th, 5th, 6th and 8th time. The colors are based on the Vaisaila RS41-SG measured altitude with the maximum altitude set to 1000 m.**

460

[Figure]

**Figure 9 : Comparison of the correlation coefficient and the linear regression coefficients between the S1H2 and the RS41-SG temperature (a), relative humidity (b), altitude (c), wind speed (d), zonal winds (e) and meridional winds (f) for all the flights where the S1H2 was used respectively for the 1$^{st}$, 2$^{nd}$, 3$^{rd}$, 4$^{th}$, 5$^{th}$, 6$^{th}$ and 8$^{th}$ time.**

---

## Author Comment (AC3) · 7 Dec 2018

We thank the anonymous referee for the helpful comments. We are responding to all the comments of the referee #2 in the supplement and we have prepared a revised manuscript where changes are marked and removed parts as follows: blue correspond to referee #1 RC2, red referee#1 RC3, green referee #2 RC4.

Please also note the supplement to this comment: https://www.atmos-meas-tech-discuss.net/amt-2018-179/amt-2018-179-AC3-supplement.zip

---

## Author Comment (AC4) · 7 Dec 2018

We thank the anonymous referee for the quick review.

———————————————————

---

## Author Response (AR1)

**Final answer for AMT-2018-179**

**Title:** Evaluation of Windsond S1H2 performance in Kumasi during the 2016 DACCIWA field campaign

Authors: Geoffrey E.Q. Bessardon, Kwabena Fosu-Amankwah, Anders Petersson, Barbara J. Brooks

We thank the anonymous referee for the helpful comments. We are responding to all the comments of the reviewers in this document and we have prepared a revised manuscripts where changes are marked and removed parts as follows: red correspond to reviewer #1 RC2, blue reviewer #1 RC3, green reviewer #2 RC4. In the following, comments of the reviewers are given in bold and italic with our responses are given in normal font. The final answer is structured as follow: 1) answer to reviewer #1 RC2 comment, 2) answer to reviewer #1 RC3 comment, 3) answer to reviewer #2 RC4 comments.

**1. Answer to referee #1 RC2**

**Overall comments:**

This paper describes progress towards developing a less expensive but reliable upper-air radiosonde. To evaluate their newly developed S1H2 sonde they compare its data to observations from high-quality Vaisala RS41-SG sondes. The observations were taken from 33 launches during the DACCIWA field campaign in Western Africa. Basically the authors conclude that the S1H2 sonde is a work in progress with the main issues being the poor performance of the GPS sensor leading to questionable winds and the slow response time of the temperature and humidity sensors. It's ironic that an instrument called a "windsond" would do such a poor job measuring winds. They conclude by offering some recommendations for future improvements. From the limited comparisons shown between RS41 and S1H2 observations, it is hard to properly judge the performance of the windsond. For example, only one intercomparison flight is made for data extending above the boundary layer. Figures 5-7 show data from this one flight. To get meaningful statistics to evaluate the windsond, data from 20 or more flights should be presented as in Jensen et al. (2016) and similar intercomparison studies. For soundings within the boundary layer, analyses are shown from (I believe) eleven flights (Figs. 8-9) and in a format that is difficult to interpret. I would recommend that analyses be presented in a more conventional format as biases and rms differences between the RS41 and windsond (see Fig. 8 of Jensen et al. 2016). While the paper has some major concerns in the way the analyses are presented, it is still of value in that it is introducing a new instrument with a promising upside that is in the early stages of development. Under major comments below I suggest several areas where paper could be improved.

**Answer:**

We agree that only one sounding over the boundary layer is not statistically sufficient to assess the S1H2 performances. A key point about the S1H2 is that its re-usable this particular sonde is of interest as as it illustrates the reusable capacity. Launches in which an S1H2 and RS92 are tapped together by default result in the loss of the sonde and so analysis of its re-use performance cannot be undertaken. Eleven RS41-SG have been launched simultaneously with two S1H2 during the re-use evaluation, so twenty-two S1H2 flights have been compared to eleven RS41-SG in the boundary layer. This method allowed us more data comparison and also less travel time to search the sondes as the two S1H2 launched simultaneously where landing in the same area.

**Major comments:**

While the windsond system is being marketed as a less expensive replacement to more conventional sondes, no where is the cost of the sonde system (laptop, antenna, etc.) and sondes mentioned in the paper. Please discuss this information.

Answer: We do not want to state the price of the sonde in the paper as the price is subject to evolution. This sondes requires a smaller ballon and and consequently less helium so saving are made on this side. The sonde is also re-usable so re-using the sonde up to 8 times can constitute a significant saving.

**Line 24: The vertical resolution is also a function of the sampling rate.**

Answer: True it has been added

Line 28-33: So the US sites are spending ~\$237K per site per year. I would assume that the US sites are some of the more costly ones to maintain around the globe so I would guess your \$440M is gross overestimate. You might want to state a range like from \$237M to \$440M. The statement referencing Martinez (2016) is confusing. It reads as if you saying that Greenland has 40 operational sites? I'm assuming you mean the Arctic has 40 sites. You may want to reword this statement. Also, is Martinez (2016) a valid reference?

Answer: A sentence to clarify that this estimate is only valid for the US has been added, the reference to Martinez has been removed as it was confusing and does not add essential information in this paper. Some corrections in the last few sentences of the introduction have been made to clarify that Windsond is a less expensive (in terms of initial set up and consumable costs) alternative for boundary layer radiosounding

With an operational ceiling of 6 km, it does not seem that the windsond system can be used to replace the sondes currently being used at operational sites which record data to 25 km and higher. With this mind what are the practical research applications of the windsond S1H2 as an upper-air system? Because of its limited range it seems best suited for use in boundary layer studies. However boundary layers are often characterizes with sharp gradients in potential temperature and moisture which the S1H2 has difficulty resolving because of its slow response time. Please discuss. Are there plans to use improved T and RH sensors with a better response time?

Answer: Windsond main objective is to enable boundary layer studies, so the Windsond has no upper-atmosphere application. The current response time limitation is the weakness of the system for boundary layer applications. In small scale, Sparv Embedded uses temperature and relative humidity sensors with a better response time, but currently, the cost is high in the context of radiosondes. Lowering the per-unit cost would take a sizeable investment in the production process to automate assembly and calibration. A key point is that the windsond system can be used in

countries with limited resources to deploy a radiosounding network utilising the more accurate but more expensive sondes as well as field campaigns were multiple shallow sounding are required. An example of this application is the VORTEX-SE project, where Penn State University released 24 sondes at the same time to study winds around storm supercells and might release as many as 100 at a time in the next season. This is a unique feature of Windsond for dense measurements (http://windsond.com/swarmsonde-is-in-the-news/)

**Line 56: Why is the operational ceiling at 8 km? Is this the burst altitude of the party balloon used with the sonde or are there some other considerations?**

Answer: Supporting soundings higher than 8-10 km requires technology that more closely resembles traditional radiosondes, diminishing the advantages of Windsond. The sondes would be heavier and require a more expensive sensor suit to overcome the hasher measurement conditions in the upper atmosphere. Moreover, while not all users find it worthwhile to recover the sondes, at high altitude the sondes would drift too far for any recovery to be feasible. Windsond does not try to replace traditional sondes, but rather enable new low-altitude soundings.

**Figure 4: It's difficult to see the ruler in this picture to get an idea of the length of the sonde.**

Answer: The picture brightness has been fixed to see the ruler.

**Line 104: Also mention that the RS41-SG pressure calculation uses the hypsometric equation.**

Answer: This has been added line 104.

**Line 123-124: Please clarify what it means "that the MW41 only produces the highest degree of signal processing". In other places you mentioned RS41 data before and after processing.**

Answer: The predecessor of the MW41, the MW31 had a research mode and an operational mode. The research mode processes the data as little as possible only correcting solar radiation and pendulum effects, while the operational mode produces the highest degree of signal processing filtering raw data and interpolating discontinuous data. The MW41 has only the operational mode available, to obtain the equivalent of the MW31 research mode data (data before processing in the text) the flight have to be simulated from the flight archive with the minimum amount of data processing enabled

**Line 126: Please clarify what corrections have been introduced. Have these corrections been implemented in the results from this study?**

Answer: The sentence has been changed to: "During this experiment, the uncorrected data have been used, but the ground pressure altitude and temperature have been adjusted to the value measured by the ground-based instrumentation available on the Kumasi supersite."

**Line 153: What is experiment 6?**

Answer: The experiment 6 is the reproducibility experiment presented in section 6 this information has been added to the text

**Line 167-168: This discrepancy between sensors at 2000 m is difficult to see in the manner that the data is displayed. Could the data be presented as a function of height or pressure to better show this?**

Answer: We have chosen to directly compare each variable as the altitude error on the Windsond S1H2 would have superposed on each sensor error and not each sensor performance. Moreover, the Vaisala system does not have a pressure sensor and the pressure is calculated by the MW41 as detailed in section 3.3 while the Windsond has a pressure sensor so the profiles as a function of pressure would display data as a function of a calculated variable in one hand to a measured variable on the other hand. We could also display both sonde data as a function of the Vaisala altitude, but this will involve modifying the shape of the S1H2 profile to fit in the Vaisala altitude profile and consequently not display the actual profile obtained with the S1H2 system. We consequently think that displaying the data of the Vaiala sonde as the function of the Windsond data is the best way to assess the performance of each sensor without interference from other sensor errors.

**Line 176: Please verify that Vaisala does not use GPS differential correction to compute winds as I thought they did. In fact this statement seems to contradict what is said earlier in lines 111-113. Did you mean the S1H2 does not do differential correction to compute winds.**

Answer: Section 3.4 lines 110-118 have been rephrased for clarity as it was confusing the way it was presented.

The Vaisala sonde uses differential correction for latitude longitude and altitude positioning. However, the Vaisala system computes the wind speed independently from the position using the GPS signal without Differential correction.

The Windsond system does not have a differential correction on its GPS to compute latitude and longitude and uses pressure to compute altitude.

Line 176: It seems really puzzling why the Windsond winds are of such poor quality. For example the IMET sonde system does not use a differential wind correction and its winds compare quite favorable to the RS41 sonde. Can you give some explanation for the poor performance of the Windsond winds? Is some of this error due to the pendulum motion of the sonde swinging below the balloon which is filtered out in the RS41 processing but not filtered out by S1H2 system?

Answer: The poor agreement surprised Windsond as informal comparisons with Vaisala and Graw have shown good agreement in wind speed and direction. The pendulum is a possibility as the

Windsond has since increased the length of the tether line. During the performance flight, both Windsond and Vaisala were on the same tether line, while on the reproducibility flight the Windsond was on a shorter line compared to the Vaisala and there was no significant increase in the wind speed and direction error between the two experiments. This suggests that the pendulum correction does not have a significant impact on the wind speed and direction.

Wind gusts and local wind variation associated with the general slower response time of the Windsond system are more likely to explain this error.

**Line 195: One sounding does not provide statistically significant evidence for this statement. See comments above.**

Answer: We agree that one flight is not statistically significant for definitive conclusion. We have added that this has to be confirmed by more flights.

**Section 5.2.2. So to clarify are you saying that the results shown for the S1H2 have no post processing and no corrections applied? Can you state what processing and corrections the MW41 performs. You mention smoothing in line 194. Is this smoothing of all fields? Is the balloon pendulum motion only taken out in the MW41 processed data?**

Answer: The S1H2 has no post-processing applied especially no pendulum and radiation correction while the data processed by the MW41 have been filtered, pendulum and solar radiation effect have been corrected, and data gaps have been interpolated.

**Figure 7: It appears that the surface or starting pressure used is different between the systems. Why is this?**

Answer: The surface pressure is the same there is one point with the coordinate (0,0). However it is hard to see so the 2 zero lines have been added to figure 7 for clarity.

**Line 207: Does the pressure difference between the two systems continue to increase with altitude?**

Answer: A typo has been found and has been corrected in the manuscript. The altitude error increases with height while the pressure error remains stable.

**Line 229: What is a .kml file? Does this need to be mentioned?**

Answer: kml files or Keyhole Markup Language files are files used for expressing geographic annotation and visualization within Internet-based, two-dimensional maps and three-dimensional Earth browsers such as Google earth.

As this information is not essential the information have been deleted and replaced with: "the system automatically predicts and displays the expected landing point on a map view."

**Line 232: Are these flashes of light coming from the sonde? Please clarify.**

Answer: Yes, the flashes of light are coming from inside the sonde we changed the sentence to "the contact between the sonde and the ground station was established, the sonde started immediately to emit loud beeps (about 15 seconds time interval) and flashes of light."

**Line 235: Have you considered if a 4m string is long enough to prevent balloon effects on the sonde observations? I believe the Vaisala system uses a much longer string (20-30m) to prevent any balloon impacts on the sonde data.**

Answer: The 4 m string has been chosen following the constructor recommendation, however Windsond has since changed its recommendation for the sondes suggesting that balloon effects have been noticed.

During the performance flight, both sondes where tapped together under the 20 m meter string and the winds errors are a similar magnitude as during the reproducibility experiment so the balloon effect does not seem to have a significant impact on the sonde data.

**Line 244: Please clarify what the "data alteration study" is.**

Answer: The data alteration study is the study of the alteration of the sounding performance through sonde re-use. The text was changed to "data alteration from sonde re-use study"

**Line 285: This is good suggestion and should be a standard practice for all flights (i.e., proper surface base-lining of sondes)**

Answer: Agreed

**Table 3: Please mention the RH sensor response time.**

Answer: The response time of the RH sensor was added

**Listed below are some additional minor suggested changes the authors may want to consider.**

**Minor comments**

All the suggested rewording have been applied to the manuscript

Line 48: suggested rewording, "because the LLC cover …" Line 50: suggested rewording, "boundary layer sounding during …" Line 69: "Figure 4 shows the Windsond …" Line 74 "sensor is used in …" Line 134 and elsewhere like Table 6: mention if time is GMT or LT. Line 160: "all the assessed meteorological parameters …" Line 168: "sudden warming …" Line 171 and 172: change "reply" to "response" Line 234: "When re-using the sonde …" Line 256: "for locating soundings …" Line 289: Seems like "different altitudes" should be "lower altitudes". This would be a good place to state the specific niche that the Windsond is trying to fill. Certainly in its current configuration it will never be used as an operational sounding. Line 292: "longer response time …"

**2. Answer to referee #1 RC3 for AMT-2018-179**

Overall comments: The authors have offered adequate responses either by addressing my comments or revisions to the paper. However it should be clearly stated in the abstract that this Windsond is intended primarily for collecting boundary layer observations.

Answer: This information has been added on the first sentence of the abstract

Also note that boundary layers are typically 500 m over the tropical oceans but can be 5 km deep under summertime continental conditions. So in the first sentence of the conclusions where you state that it measures conditions at lower altitudes, lists an approximate height range where observations are considered good. For example,"... lower altitudes (up to 2 km)" or whatever altitude you trust your data.

Answer: This information has been added to the first line of conclusion

**Finally, in your response you mention that you thought the balloon did not effect the winds. But there is also a concern during daylight flights that radiative effects off the balloon with a short 4m string could effect the T and RH measured by the sonde.**

Answer: That is true, our answer was focussed on the wind speed error as the wind speed error was the largest. The balloon used under a 4 meter rope for the reproducibility experiment was smaller than the balloon used during the performance flight when the Windsond was taped to Vaisala sondes under the 20 meter string. The T and RH errors during the performance flight are a similar magnitude during the reproducibility experiment so the smaller ballon under a 4 m rope does not seem to have a similar impact on data compared to the larger balloon with a 20 m rope. However, we recognize that the use of a longer string with the smaller balloon would be an inexpensive way to reduce the radiative effects on the data collected by the Windsond system.

**Suggested rewording:**

All the suggested rewording have been applied to the manuscript

Line 30: This rough estimate varies regionally as the price of labor, helium and balloons is not the same around the globe. Yet operational costs are a significant investment in countries with limited resources.

Line 111-115: "... the Vaisala ground station has a GPS receiver ... However, wind speed and direction are determined independently from the GPS position using the GPS doppler frequency shifts.

Line 117: "Similar to the RS41-SG ..."

Line 206: "... performed. To be statically significant this result needs to be verified with additional performance ..."

*Line 239: "During the descent after the sonde loses contact ..."*

**3. Answer to referee #2 RC4 for AMT-2018-179**

This paper presents an evaluation of a relatively new low-cost radiosonde system against a well-established and widely used radiosonde based on measurements performed in June and July 2016 during a field campaign in Ghana - Western Africa. The low-cost radiosondes were recovered by the operators and reused up to 8 times, which allows the authors to analyse a relatively high number of ascends. It is shown that under "simple" atmospheric conditions temperature, humidity and pressure measured by both systems compare reasonably well, but as soon as larger vertical inhomogeneities occur the lowcost radiosonde suffers from slow sensor response and hysteresis. GPS-derived wind from the low-cost system is of very bad quality.

Unfortunately, the paper suffers from several weaknesses starting by the design of the measurements, missing technical information, lack of measurements under laboratory conditions and a very limited analysis of the data. The authors miss to cite and discuss relevant literature e.g. Legain et al. 2013 doi:10.5194/amtd-6-3339-2013 and Nash et al, 2010 WMO Report No. 107 Instruments and Observations. The weather situation is not sufficiently discussed and taken into account. Overall it seems to me that the paper is a kind of side product produced with minimal effort.

I think that the paper will not warrant publication as long as a mayor revision is done which addresses the following comments.

We would like to thank the reviewer for the references to the relevant literature we have missed and the their comments concerning the contextualisation of the work. This work was made in the context of the DACCIWA field campaign were the Windsond S1H2 was being integrated into a large scale scientific sounding programme for the first time. This sonde has never been used in this manner before hence we took the opportunity presented by the DACCIWA field campaign to compare the radiosonde with a proven system: providing benchmarking for the interpretation S1H2 sonde data obtained as part of DACCIWA. The experimental design was limited by the needs of the field campaign and the resources available but despite these limitations it did allow the identification of a number of issues with the Windsond S1H2 and to feed these back to the manufacturer to help the development of a reusable sonde system that is easier to use than the system presented by Legain et al., 2013.

The conclusion drawn in this paper is by no means a definitive conclusion on the Sparv Embedded system but a list of recommendation for development as well as recommendation for the future users of the DACCIWA field campaign data and as such is still usefull for the community.

**Specific comments:2**

Page 2 The first section is a marketing analysis which is mostly irrelevant if you want to discuss a reusable low cost sonde that is limited to 6000 m altitude. Sonde costs are fixed - price differences for launches in different regions depend on logistics and local labour.

Answer: Agreed, following reviewer #1 comments we have noticed that the first section is confusing so we added "This rough estimate varies regionally as the price of labour, helium and balloons and is

not the same around the globe. Yet operational costs are a significant investment in countries with limited resources."

We have introduced Legain et al., 2013 system and discussed the limitations of the system for the development of an operational network using this sonde: "Re-usable sondes have been introduced for the first time by Legain, et al., 2013 which modified a Vaisala sonde to enclose it in a cage which is tied to a couple of balloon. The caged allowed the balloon to detached at a desired altitude and slowly descend before recovery. Despite this system has shown successful results in pressure temperature and humidity, and recovery rate it does not asses the effect of the cage and the two balloons on the obtained wind profile. The sonde modification required makes the use of this system more complex and can be an obstacle towards a global use of the system, this shows that re-usable sonde technologies are still a work in progress where manufacturers can develop their own solutions."

If the sounding program had the objective to evaluate the Windsond performance already from the beginning please explain the following: 1) Why is there only one tandem flight reaching higher altitudes performed 2) Why are all low altitude intercomparison flights performed only at 0600 and not distributed over day and night or at least over the launch times shown on figure 2. As the sondes were recovered no significant additional costs would have been created. 3) Why are the RS42 and Windsond not tied together for the low altitude intercomparison flights – the resulting spacial difference makes it impossible to separate instrument errors from atmospheric variability.

Answer: The goal of the DACCIWA ground field campaign was to provide a high-quality comprehensive dataset for processes studies, in particular interactions between low-level clouds (LLCs) and boundary layer conditions. The DACCIWA radio sounding program was then designed to complete these main objectives. The Vaisala RS42s were launched to provide synoptic observations during the campaign with complementary synoptic measurements during IOPs. The Windsond S1H2s were launched to provide more frequent boundary layer sounding during DACCIWA IOPs, to observe the evolution of the LLCs, and associated phenomena such as the Nocturnal Low-Level Jet (NLLJ) The frequent radiosounding program thus focussed on night-time measurements as detailed on figure 2.

As the S1H2 was never used in the context of a field campaign we had to control the quality of the S1H2 in order to facilitate the interpretation of the data recorded by the S1H2. This performance assessment had however to be done without impacting on the main objectives. We agree that only one sounding reaching higher altitudes is not statistically sufficient to assess the S1H2 performances. Launches in which an S1H2 and RS92 are tapped together by default result in the loss of the sonde and so would compromise the completion of the Windsond objectives. We were planning to perform more intercomparison flights toward the end of the campaign unfortunately, as quoted in section 6.2 the radio receiver has been damaged during the campaign preventing us from completing this final objective.

Due to limited human resources (5 scientists separated into 2 teams performing 12 hours shifts onsite to run the whole Kumasi supersite instrumentation), distributed flights over day and night were not possible, thus to comply with the DACCIWA objective the frequent radiosounding program focussed on night-time measurements as detailed on figure 2. It was advised by our Ghanaian partners to avoid going out of the supersite at night to avoid encounters with tropical wilderness and limit the robbery risk. For these reasons, sonde recovery took place after sunrise, thus, to limit the time between the launch of the sonde and the recovery, all the frequent radiosounding flights took place in the last part of the night and the intercomparison flights at 0600 UTC.

According to Sparv Embedded, the Windsond S1H2 system is reusable, requires a smaller balloon and less helium and can receive multiple sonde, for the first utilisation of the sonde these features have to be tested. The S1H2 sonde were launched using manufactuer recommendations to evaluate the performance of the sonde in regular use. At low wind speeds the signal-to-noise ratio in the wind speed measurement is worse, so reducing the ascent speed can adversely affect the wind speed accuracy. For these reasons the sonde were not tied together for the low-altitude comparison flights. We agree that this experimental design does not allow us to quantitatively asses the Windsond S1H2 performance, however, this design allows us to qualitatively asses the system where the wind speed and direction issues have been confirmed and evaluate an eventual data alteration trend through use.

Eleven RS41-SG have been launched simultaneously with two S1H2 during the re-use evaluation, so twenty-two S1H2 flights have been compared to eleven RS41-SG in the boundary layer. This method allowed us more data comparison and also less travel time to search the sondes as the two S1H2 launched simultaneously where landing in the same area.

Page 3 Please us UTC or LT but not AM / PM Is Fig. 2 really needed ? Please explain what you mean with simultaneous launched (see above). Please give more information about the calibration of the Windsond. Do sondes have individual factory calibrations stored on the sonde or does the manufacturer rely on the quality of its sensors only ? How is the multi sonde reception realized – please give details on the receiver technology. Please use Kelvin instead of \_C for Accuracy and Resolution in Table 2

Answer: Following reviewer #1 recommendation all the times are now changed to UTC. Individual calibration is done by the sensor manufacturers, thus stored in the sensors, multi sonde reception is realized by time-division multiplexing. As requested temperatures are now expressed in Kelvin in Table 2.

**Table 3-5 Anders Petersson is affiliated to the manufacturer of Windsond. You should be able to give detailed information about the sensors used in Windsond and their performance instead of "not available (to be assessed)". Is the given value for pressure accuracy valid for Vaisala or Windsond? Why is the Wind speed accuracy relative to the wind speed?**

Answer: The cell alignment for the table 2 leads to a confusion for the pressure accuracy, the given accuracy value is only valid for Windsond, the Vasaila value is defined as combined uncertaincy and reproducibility. At low-speed the signal-to-noise ratio in the wind speed is worse, thus the dependency of wind speed accuracy relative to wind speed.

**Page 5: Pressure sections: Please include in the discussion the results of the WMO radiosonde intercomparison 2010 about direct pressure measurements vs. derived pressure.**

Answer: We have added: "The WMO radiosonde intercomparison experiment 2010 showed that pressure measurement derived from geopotential heights and radiosonde measurements of temperature and relative humidity profile were very reproducible and suitable for all radiosounding operations for system where GPS system are set up correctly which includes the Vaisala system. This shows that the Vaisala derived pressure is a reliable reference to assess the Windsond pressure sensor, and the Windsond cost can be lowered by removing the pressure sensor in future version of the Windsond system depending on its GPS system accuracy."

Page 6: Please explain uncorrected data vs data correction for all parameters for the Windsond. What was the procedure to "adjust" ground pressure altitude and temperature? Wow large were these "adjustments" and why was this not done for humidity? I am still astonished that only one tandem flight to higher altitudes was performed! A larger number of such flights under different weather situations as well as during day and night would have improved the evaluation significantly. The flight was in 2016 and not 2006. Since all flights were performed during night or early morning radiative effects cannot be evaluated. Experimental design needs to be explained in more detail.

What was the length of the line connecting the sondes to the balloon? How did you tape the sondes together? Is it excluded that waste heat of one sonde influenced the other? Why did you set the Windsond acquisition to 3 seconds - according to table 1 the measurement cycle is 1s for both sondes. Please give details about the weather situation.

Answer: As mentioned in section 4 the Windsond S1H2 firmware has a single operational mode and produces uncorrected data, the only correction applied was to simply differentiate the ground value from the ground-based instrumentation and apply this difference to the profile. The altitude correction was in the [-10; +10] m range, pressure correction was [-2;+2] hPa, and temperature [-2;+2] range. In the stable nocturnal boundary layer, surrounding vegetation is expected to affect the local humidity values while having a limited effect on the temperature, for this reason humidity correction were not performed.

We agree that a larger number of flights would have improved the evaluation, however, the goal of this evaluation was to assess the quality of the Windsond data in context of the DACCIWA data analysis framework and also to address limitations of the Windsond system in the tested configuration. The radiative effect could not be be evaluated but needs to be evaluated for future use of the system. The flight was in 2016 we have corrected the typo.

The line connecting the sonde to the balloon was 20 m and the sonde were taped together making sure that the temperature and humidity sensors of each sonde were not interfering or influenced each other. At the time of the test, 1 s sampling rate was not compatible with the version of the firmware tested this is now available in the new version.

5.1.2 Should be renamed to Signal processing for low altitudes – Boundary layer higght was not detected - I would expect a boundary layer height around 100 m at the launch time of the sonde rising up to 1500 m during the day in this region during the monsoon period.

Answer: Agreed the title has been modified

Page 7: Can you explain why you have chosen different ascent rates and non-attached sondes for evaluating the reproducibility? I can't see any sense in this procedure since natural atmospheric variation will be at least in the same range as the instrument error. Profile comparison – It would be nice to have a profile plot if you do profile comparisons! Instead of showing scatter plots it would make much more sense to plot vertical profiles of PTH as well as wind for both sondes with an additional profile showing the vertical profile of the difference (Vaisala – Windsond) for each parameter together with the accuracy as stated by the manufacturer's datasheet. This would allow a meteorological interpretation. How do you measure cloud top temperature above the cloud top – The RS41-SG sensors are detecting the cloud top temperature and humidity before the S1H2 : : ...????

Answer: As mentioned in our answer concerning the page 2 comment we have chosen a different ascent rate in order to test the accuracy of the sonde by following constructor recommendation, test the multisonde reception and recovery system, and to increase the number of sonde tested.

We have chosen to directly compare each variable as the altitude error on the Windsond S1H2 would have superposed on each sensor error and not each sensor performance. Moreover, the Vaisala system does not have a pressure sensor and the pressure is calculated by the MW41 as detailed in section 3.3 while the Windsond has a pressure sensor so the profiles as a function of pressure would display data as a function of a calculated variable in one hand to a measured variable on the other hand. As discussed in the WMO intercomparison, basic raw data are to diagnose problems with a radiosonde during evaluation. We could also display both sonde data as a function of the Vaisala altitude, but this will involve modifying the shape of the S1H2 profile to fit in the Vaisala altitude profile and consequently not display the actual profile obtained with the S1H2 system. We consequently think that displaying the data of the Vaiala sonde as the function of the Windsond data is the best way to assess the performance of each sensor without interference from other sensor errors.

The structure of this sentence was confusing and has been changed to: "For both temperature and relative humidity, the RS41-SG sensors are detecting the sudden temperature and humidity changes consecutive of the top of a cloud before the S1H2 sensors"

**Page 8: Change reply time to response time**

The atmosphere is characterized by vertical inhomogeneities, inversions and clouds – radiosondes therefore have to have sensors with low response time and neglectable

**hysteresis – if this is not the case the sonde is simply not suitable as radiosonde – or only for nice weather well mixed cloud free boundary layer.**

Answer: Following reviewer #1 comments answer time has been replaced by response time.

The current response time limitation is the weakness of the system for boundary layer applications. In small-scale, Sparv Embedded uses temperature and relative humidity sensors with a better response time, but currently, the cost is high in the context of radiosondes. Lowering the per-unit cost would take a sizeable investment in the production process to automate assembly and calibration.

Page 9: More recent versions of the Windsond firmware certainly correct the altitude bias - have you checked this? Is it possible to reprocess the measurements to verify? To me it is not shown that newer firmware versions correct the altitude bias. The conclusions are too favourable – Windsond cannot handle inhomogeneities due to the high response time of the sensors, GPS derived wind error is far above the 5% error given by the manufacturer and to my opinion useless. It is not shown that at least the altitude correction in the latest versions of Windsond improve the systematic altitude error. As the WMO intercomparison results and the Vaisala sonde show pressure sensors are not needed any more for radiosondes – the "robust performance" of the pressure sensor us unfortunately only of minor importance.

Answer: We agree the word certainly was too favourable for something we have not tested, this has been corrected to probably. We have added the statement in the conclusion : "These limitations make the deployment of an operational network using this system under the tested configuration impossible."

The WMO intercomparison shows that pressure sensor are not needed anymore for radiosondes in situation where GPS radiosondes are set up correctly which is not the case of the Windsond. Thus the evaluation of the pressure sensor is important to assess the use of the Windsond S1H2 data in a meteorological context such as tephigrams.

**Page 10:**

**The experimental design shows several weaknesses – as already addressed the fact that the sondes were not tied together during the ascends makes it nearly impossible to separate instrument error and atmospheric variance. I would recommend to test each sonde prior nest launch instead of a simple visual inspection.**

Answer: In the morning stable boundary layer horizontal variations are small so the instrument error would still be significant compared to the atmospheric variance. We agree, however, that atmospheric variance will add some noise to the error recorded between sondes. Despite these weakness this design allows us to show that there is no clear trend in data alteration consecutive with sonde re-use during the experiment.

We agree that testing sonde prior the next launch should be the standard, however limitated ressources made a detailed inspection impossible. This standard is unfortunately not always respected as Legain et al, 2013 also relaunched a large number of sonde immediately after recovery.

I would strongly recommend to perform additional measurements with a larger number of sondes under laboratory conditions to determine sensor accuracy and inertia over a wide range of temperature and humidity and to compare the results to the sondes datasheets first. Reproducibility can also better be tested in a combination of repeated tandem flights and climate chamber measurements – this would allow the separation of sensor degradation and atmospheric influence in real atmospheric conditions.

Answer: These are interesting comments, but these go beyond the scope of this paper, here we test the performance of the sonde during a field campaign which is similar to Legain et al., 2013 where the system was tested during 2 field campaigns. The sonde recovery system cannot be tested in an atmospheric chamber as well as the different natural hazard encountered in a rough environment such as West Africa. Despite the different limitations of the experimental design we have been able to identify some limitations of the system especially for the GPS system which are worth publishing to provide indication to Sparv Embedded and future users of the system.

**Table 6 is unreadable – it extends 4 pages – please consider a condensed way of presentation.**

Answer: Agreed, we have substituted table 6 with figure 8 that condenses all the information of table 6

|         |             |             |             |             |             |             |             |                 |             | Suc
Suc
Suc | ccesfu
ccesfu
ccesfu
ccesfu | Il Yes
Il Yes
Il No R
Il No R | Recov
Recov
lecove | ery Ye
ery Ne
ery Ye
ery No | 25
0
5 |                 |             |             |             |             |                 |  <li>te</li> <li>p</li> <li>1</li> <li>2</li> <li>2</li>  | est
erforn
S1H2
S1H2
S1H2
S1H2 | nance
+ RS | 41          |             |             |                |             |             |             |             |             |
|---------|-------------|-------------|-------------|-------------|-------------|-------------|-------------|-----------------|-------------|-------------------|--------------------------------------|----------------------------------------|--------------------------|--------------------------------------|--------------|-----------------|-------------|-------------|-------------|-------------|-----------------|--------------------------------------------------------------------|-----------------------------------------------|---------------|-------------|-------------|-------------|----------------|-------------|-------------|-------------|-------------|-------------|
| Sonde B |             |             | 470
(1)  |             | 466
(1)  |             | 376
(1)  |                 |             |                   | 468
(2)                           |                                        |                          | 376
(3)                           |              |                 | 472
(1)  |             |             | 472
(2)  |                 |                                                                    | 464
(6)                                    |               |             | 472
(4)  |             | 472
(5)     | 374
(2)  | 346
(1)  | 374
(3)  | 346
(2)  | 386
(1)  |
| Sonde A | 471
(1)  | 471
(1)  | 343
(1)  | 465
(1)  | 335
(1)  | 343
(1)  | 464
(1)  | 465
(2)
¥ | 305
(1)  | 335
(1)        | 467
(1)                           | 468
(1)                             | 464
(1)               | 468
(3)                           | 355
(1)   | 305
(1)
¥ | 468
(4)  | 464
(2)  | 467
(1)  | 468
(5)  | 376
(2)
¥ | 467
(2)                                                         | 472
(3)                                    | 468
(3)    | 464
(3)  | 411
(2)  | 376
(4)  | 468
(8)
 | 411
(3)  | 382
(1)  | 472
(6)  | 382
(2)  | 469
(1)  |
|         | 10/06 13:09 | 18/06 03:23 | 18/06 05:46 | 21/06 02:43 | 21/06 08:49 | 26/06 03:00 | 26/06 05:51 | 26/06 08:36     | 28/06 05:44 | 29/06 02:44       | 29/06 05:46                          | 29/06 08:43                            | 01/07 02:37              | 01/07 05:21                          | 01/07 08:35  | 03/07 02:49     | 03/07 05:41 | 03/07 08:44 | 08/07 02:39 | 08/07 05:44 | 08/07 08:40     | 11/07 02:48                                                        | 11/07 05:46                                   | 11/07 08:40   | 14/07 02:59 | 14/07 05:46 | 14/07 08:47 | 18/07 05:54    | 18/07 08:50 | 21/07 05:55 | 21/07 09:13 | 24/07 05:48 | 24/07 09:23 |

Figure 1 Timeline listing sounding time in UTC, the shapes indicate the corresponding number of radiosonde S1H2 launched (test denotes the test sonde, performance denotes the S1H2 launched taped to an RS41-SG, +RS41 denotes simultaneous launched with the Kumasi Agromet supersite), the sonde id with the number of time the sonde has been used under brackets, the colors indicates flight result and the recovery result.

**Page 11:**

Please give the percentage of unsuccessful flights and flights with sondes that did not cut off. Is the number of data from sondes that did not cut off large enough to do a representative evaluation for altitudes between 650 and 1000 m?

**It is nearly impossible to separate the different markers in Fig. 8. Maybe separated figures would help.**

Answer: Only 3 flights with failed cut-off were launched simultaneously with an RS41 so an evaluation between 650 and 1000 m would not lead to a statistically representative evaluation of the Windsond system. This evaluation would also be beyond the scope of this evaluation which focusses on how well the low-level cloud and low-level jets are represented in this study.

We agree that on figure 8 the markers are nearly impossible to separate, but separated figure would increase the number of figures without leading to more interesting conclusions.

As you have a large number of flights over several days available I would recommend to do not only a statistical analysis based on scatter plots and regressions but also a more meteorological where you create classes of different weather situations e.g. with and without low level clouds and analyse the behaviour of the sondes along the vertical profile.

Answer: During the DACCIWA field campaign the low-level cloud was a recurrent feature and only one IOP night was identified without a low-level cloud so a meteorological climatology based analysis would not be relevant.

**Fig. 9 – why do you use lines to connect the markers?**

Answer: We decided to use lines to connect the marker to help the reader to see that there is no real trend between sonde usage and data alteration.

**Page 12: A check of the sonde sensors before reusing it should be the standard procedure – see my comment to page 10.**

Answer: Agreed this should be a standard but field campaign constraints can limit the time on the sonde check to avoid the radiosonding program to compromise other instruments objectives.

A system for low altitude rapid soundings using high quality radiosondes was already introduced and tested by Legain et al. 2013. The questions to me is if a low cost and unfortunately low-quality system like the Windsond really makes sense with all the weaknesses we have seen in your evaluation especially - considering the fact that higher quality sondes can also be recovered and reused so that the cost difference between the sondes gets even less important. Please discuss!

Answer: As remarked by the referee #1 our conclusions shows that the Windsond S1H2 is a work in progress and the result presented in this manuscript are introducing a new instrument in the early stage of its development. The system presented by Legain et al 2013 does not asses the consequences of the drag generated by the protection cage on the measured wind profile. Moreover, the modification applied to the sonde system required qualified personal and can limit the generalised used of this sonde.

A key point is the Windsond system does not requires any modification or complex balloon system so if that system become accurate enough it will provide a easy to use solution in countries with limited to deploy a radiosounding network utilising the more accurate but more expensive sondes. The multi sonde capability is also another key point for field campaigns were multiple shallow sounding are required. An example of this application is the VORTEX-SE project, where Penn State University released 24 sondes at the same time to study winds around storm supercells and might release as many as 100 at a time in the next season. This is a unique feature of Windsond for dense measurements (http://windsond.com/swarmsonde-is-in-the-news/).

**Please change longer answer time to response time!**

Change from answer time to response time have been made following remarks from reviewer #1.

It would be nice to know if newer Windsond firmware really has corrected the problems with GPS derived wind and pressure. I therefore strongly encourage you to perform the further performance evaluations and include their results into a revised version of your manuscript.

Answer: We agree that it would be interesting but this analysis would go beyond the scope of this paper and would require more time and capabilities to perform this study.

**Evaluation of Windsond S1H2 performance in Kumasi during the 2016 DACCIWA field campaign**

Geoffrey E.Q. Bessardon1, Kwabena Fosu-Amankwah2, Anders Petersson3, Barbara J. Brooks4

1 School of Earth and Environment, University of Leeds, Leeds, LS2 9JT, UK

2 Department of Physics, Kwame Nkrumah University of Science and Technology, Kumasi, Ghana

3 Sparv Embedded AB, Linköping, Sweden

4 National Center for Atmospheric Science, School of Earth and Environment, University of Leeds, Leeds, LS2 9JT, UK

Correspondence to: Geoffrey E.Q. Bessardon (eegb@leeds.ac.uk)

Abstract. Sparv Embedded, Sweden (http://windsond.com) has answered the call for less expensive but accurate reusable radiosondes by producing a reusable sonde primarily intended for boundary-layer observations collection: the Windsond S1H2. To evaluate the performance of the S1H2, in-flight comparisons between the Vaisala RS41-SG and Windsond S1H2 were performed during the Dynamics-Aerosol-Chemistry-Cloud Interactions in West Africa (DACCIWA) project (FP7/2007-2013) ground campaign at the Kumasi Agromet supersite (6°40'45.76''N, 1°33'36.50''W) inside the Kwame Nkrumah University of Science and Technology (KNUST), Ghana campus. The results suggest a good correlation between

15 the RS41-SG and S1H2 data, the main difference lying in the GPS signal processing and the humidity response time at a cloud top. Reproducibility tests show that there is no major performance degradation arising from S1H2 sonde re-use.

**1** Introduction**

5

Accurate in-situ measurements of tropospheric temperature, pressure, water vapour and wind profiles provide critical input for numerical weather forecasting and climate models, in the quantification of atmospheric thermodynamic

- 20 stability, for the development and application of remote-sensing retrievals, and as an important constraint for atmospheric process studies. Since the 1930s such measurements have been made by small instrument packages attached to balloons (Jensen et al., 2016) known as radiosondes; the vertical resolution of the profile being determined by the ascent rate of the balloon (Martin et al., 2011). The many changes in instrumentation, sounding practices and data processing are discussed at length by many authors including Haimberger 2007; Vömel et al., 2007; Haimberger et al., 2008; Rowe et al., 2008;
- 25 Sherwood et al., 2008; McCarthy et al., 2009; Miloshevich et al., 2009; Seidel et al., 2009; Dai et al., 2011; Hurst et al., 2011; Thorne et al., 2011; Moradi et al., 2013; Wang et al., 2013; Dirksen et al., 2014; Yu et al., 2015; Bodeker et al., 2016; Jensen et al., 2016.

The operational cost of launching a radiosonde is high: according to B. Blackmore 2012, personal communication, as cited by Gonzalez et al., 2012, the National Weather Service (NWS) Weather Forecasting Offices (WFO) estimates that 30 the cost per unit launch of a radiosonde in the US is US\$ 325 (Price includes radiosonde, balloon and labour) and a total of \$21,827,000 a year if 2 launches are made at 92 sites. This rough estimate varies regionally as the price of labour, helium and balloons and is not the same around the globe. Yet operational costs are a significant investment in countries with limited resources.

For many years the provision of radiosounding- technology has been dominated by the likes of Vaisala and Graw but over the last decade there has been an increase in the call for less expensive but accurate devices (Douglas, et al., 2012; Martinez 2016; Krauchi and Philipona 2016). The development of a cheaper re-usable radiosounding system could contribute to the development of a denser operational network in regions in the world with limited financial resources, as well as being useful for field campaigns where multiple shallow soundings are needed.

- Re-usable sondes have been introduced for the first time by Legain, et al., 2013 which modified a Vaisala sonde to
  enclose it in a cage which is tied to a couple of balloon. The caged allowed the balloon to detached at a desired altitude and slowly descend before recovery. Despite 
[revised manuscript text omitted]

|                             | functionality                                        |            |           |     |    |   |                 |
| Measurement Range           | 0-100% RH                                            | 0-100      | % RH      |     |    |   |                 |
| Accuracy repeatability in   | 2.0% RH                                              | 2.0 %      | RH        |     |    |   |                 |
| calibration                 |                                                      |            |           |     |    |   |                 |
| Resolution                  | 0.1 % RH                                             | 0.05 %     | % RH      |     |    |   |                 |
| Combined uncertainty in     | 4% RH                                                | Not        | Available | (to | be |   |                 |
| sounding                    |                                                      | assess     | ed)       |     |    |   |                 |
| Reproducibility in sounding | 2% RH                                                | Not        | Available | (to | be |   |                 |
|                             |                                                      | assess     | sed)      |     |    |   |                 |
| Response time (63.2%, 6 m/s | Heated sensor: <0.3 s                                | 5 s |           |     |    |   |                 |
| flow, 1000 hPa)      | Cold sensor < 10 s                         |            |           |     |    |   |                 |

| Sonde Characteristics       | RS41-SG radiosondes                 | S1H2 Windsond        |  |  |  |  |
|-----------------------------|-------------------------------------|----------------------|--|--|--|--|
| Pressure                    |                                     |                      |  |  |  |  |
| Sensor type                 | GPS-derived                         | MEMS pressure sensor |  |  |  |  |
| Range                       | Surface to 3hPa                     | 1100 - 300 hPa       |  |  |  |  |
| Accuracy                    | Defined as combined uncertainty and | 1.0 hPa              |  |  |  |  |
|                             | reproducibility                     |                      |  |  |  |  |
| Resolution                  | 0.01 hPa                            | 0.02 hPa             |  |  |  |  |
| Combined uncertainty in     | 1.0>100 hPa                         | Not Available (to be |  |  |  |  |
| sounding                    | 0.3<100 hPa                         | assessed)            |  |  |  |  |
|                             | 0.04<10 hPa                         |                      |  |  |  |  |
| Reproducibility in sounding | 0.5>100 hPa                         | Not Available (to be |  |  |  |  |
|                             | 0.2

---

## Referee Report (RR1)

**Third Review for AMT-2018-179 (Bessardon et al.)**

**Title:** Evaluation of Windsond S1H2 performance in Kumasi during the 2016 DACCIWA field campaign

**Authors:** Geoffrey E.Q. Bessardon, Kwabena Fosu-Amankwah, Anders Petersson, Barbara J. Brooks

**Overall comments:**

I agree with the other reviewer that this paper suffers from a number of critical weaknesses as was pointed out in their review. While the revisions add some additional technical information, clarify a few issues, and cite a relevant study, the majority of the shortcomings still remain (in particular, the design of the intercomparisons and lack of meaningful analysis). That being said, the paper points out some important shortcomings of the current S1H2 sonde, namely it's slow humidity response time and poor performance at computing accurate winds. Since the S1H2 sondes have been used in at least a few field programs, this information might be useful to those using the S1H2 data. So despite the obvious weaknesses of the paper, I still recommend it be published. However I do suggest that the authors include a paragraph in the summary section where they offer some practical suggestions for an improved intercomparison study to better characterize their S1H2 sonde.

**Suggested rewording:**

Line 38: "which is tethered to two balloons. Their system allowed one balloon to detach at a desired altitude and have the caged sonde slowly descend with the second balloon prior to recovery. While this system … and recovery rates, it does not assess …"

Lines 42-43: last part of this sentence is poorly written and difficult to understand.

Line 49: "in order to prepare … observations recorded" – this is difficult to understand. Are you trying to say, "in order to better understand changes in the nocturnal boundary layer, as well as …"

Line 125: Using a Windsonde without a pressure sensor requires an accurate pressure measurement at the surface if pressure above the surface is to be computed using GPS altitude information.

Line 192: "humidity changes associated with cloud top before …"

Line 312: "needs a faster response time …"

---

## Author Response (AR2)

**Answer to referee #1 third review for AMT-2018-179 (Bessardon et al. 2018)**

**Title:** Evaluation of Windsond S1H2 performance in Kumasi during the 2016 DACCIWA field campaign

**Authors:** Geoffrey E.Q. Bessardon, Kwabena Fosu-Amankwah, Anders Petersson, Barbara J. Brooks

We thank the anonymous referee for the helpful comments. We are responding to all the comments of the reviewers in this document and we have prepared a revised manuscript where major changes are marked and removed parts in red. In the following, comments of the reviewers are given in bold and italic with our responses are given in normal font.

**Overall comments:**

I agree with the other reviewer that this paper suffers from a number of critical weaknesses as was pointed out in their review. While the revisions add some additional technical information, clarify a few issues, and cite a relevant study, the majority of the shortcomings still remain (in particular, the design of the intercomparisons and lack of meaningful analysis). That being said, the paper points out some important shortcomings of the current S1H2 sonde, namely it's slow humidity response time and poor performance at computing accurate winds. Since the S1H2 sondes have been used in at least a few field programs, this information might be useful to those using the S1H2 data.

So despite the obvious weaknesses of the paper, I still recommend it be published. However I do suggest that the authors include a paragraph in the summary section where they offer some practical suggestions for an improved intercomparison study to better characterize their S1H2 sonde.

Answer: We have added a paragraph in the conclusion about practical suggestions to improve the intercomparison

**Suggested rewording:**

Answer: All the suggested rewording has been applied

Line 38: "which is tethered to two balloons. Their system allowed one balloon to detach at a desired altitude and have the caged sonde slowly descend with the second balloon prior to recovery. While this system ... and recovery rates, it does not assess ..." Lines 42-43: last part of this sentence is poorly written and difficult to understand. Line 49: "in order to prepare ... observations recorded" – this is difficult to understand. Are you trying to say, "in order to better understand changes in the nocturnal boundary layer, as well as ..." Line 125: Using a Windsonde without a pressure sensor requires an accurate pressure measurement at the surface if pressure above the surface is to be computed using GPS altitude information. Line 192: "humidity changes associated with cloud top before ..."

Line 312: "needs a faster response time ..."

**Evaluation of Windsond S1H2 performance in Kumasi during** the 2016 DACCIWA field campaign**

Geoffrey E.Q. Bessardon1, Kwabena Fosu-Amankwah2, Anders Petersson3, Barbara J. Brooks4

1 School of Earth and Environment, University of Leeds, Leeds, LS2 9JT, UK

2 Department of Physics, Kwame Nkrumah University of Science and Technology, Kumasi, Ghana

3 Sparv Embedded AB, Linköping, Sweden

4 National Center for Atmospheric Science, School of Earth and Environment, University of Leeds, Leeds, LS2 9JT. UK

Correspondence to: Geoffrey E.Q. Bessardon (eegb@leeds.ac.uk) 10

Abstract. Sparv Embedded, Sweden (http://windsond.com) has answered the call for less expensive but accurate reusable radiosondes by producing a reusable sonde primarily intended for boundary-layer observations collection: the Windsond S1H2. To evaluate the performance of the S1H2, in-flight comparisons between the Vaisala RS41-SG and Windsond S1H2 were performed during the Dynamics-Aerosol-Chemistry-Cloud Interactions in West Africa (DACCIWA) project (FP7/2007-2013) ground campaign at the Kumasi Agromet

15 supersite (6°40'45.76''N, 1°33'36.50''W) inside the Kwame Nkrumah University of Science and Technology (KNUST), Ghana campus. The results suggest a good correlation between the RS41-SG and S1H2 data, the main difference lying in the GPS signal processing and the humidity response time at a cloud top. Reproducibility tests show that there is no major performance degradation arising from S1H2 sonde re-use.

**20 **1** Introduction**

5

Accurate in-situ measurements of tropospheric temperature, pressure, water vapour and wind profiles provide critical input for numerical weather forecasting and climate models, in the quantification of atmospheric thermodynamic stability, for the development and application of remote-sensing retrievals, and as an important constraint for atmospheric process studies. Since the 1930s such measurements have been made by small instrument packages attached to balloons (Jensen et al., 2016) known as radiosondes; the vertical resolution of the profile being determined by the ascent rate of the balloon (Martin et al., 2011). The many changes in

instrumentation, sounding practices and data processing are discussed at length by many authors including

25

Haimberger 2007; Vömel et al., 2007; Haimberger et al., 2008; Rowe et al., 2008; Sherwood et al., 2008; McCarthy et al., 2009; Miloshevich et al., 2009; Seidel et al., 2009; Dai et al., 2011; Hurst et al., 2011; Thorne

30

et al., 2011; Moradi et al., 2013; Wang et al., 2013; Dirksen et al., 2014; Yu et al., 2015; Bodeker et al., 2016; Jensen et al., 2016. The operational cost of launching a radiosonde is high: according to B. Blackmore 2012, personal

communication, as cited by Gonzalez et al., 2012, the National Weather Service (NWS) Weather Forecasting Offices (WFO) estimates that the cost per unit launch of a radiosonde in the US is US\$ 325 (Price includes

35 radiosonde, balloon and labour) and a total of \$21,827,000 a year if 2 launches are made at 92 sites. This rough estimate varies regionally as the price of labour, helium and balloons and is not the same around the globe. Yet operational costs are a significant investment in countries with limited resources.

For many years the provision of radiosounding technology has been dominated by the likes of Vaisala and Graw but over the last decade there has been an increase in the call for less expensive but accurate devices (Douglas, et al., 2012; Martinez 2016; Krauchi and Philipona 2016). The development of a cheaper re-usable radiosounding system could contribute to the development of a denser operational network in regions in the world with limited financial resources, as well as being useful for field campaigns where multiple shallow soundings are needed.

Re-usable sondes have been introduced for the first time by Legain, et al., 2013 which modified a 45 Vaisala sonde to enclose it in a cage which is tied to a coupletethered to two of balloons. The caged system allowed the one balloon to detached 
[revised manuscript text omitted]

|                             | heating functionality                        |                      |  |
| Measurement Range           | 0-100% RH                                    | 0-100% RH            |  |
| Accuracy repeatability in   | 2.0% RH                                      | 2.0 % RH             |  |
| calibration                 |                                              |                      |  |
| Resolution                  | 0.1 % RH                                     | 0.05 % RH            |  |
| Combined uncertainty in     | 4% RH                                        | Not Available (to be |  |
| sounding                    |                                              | assessed)            |  |
| Reproducibility in sounding | 2% RH                                        | Not Available (to be |  |
|                             |                                              | assessed)            |  |
| Response time (63.2%, 6 m/s | Heated sensor: <0.3 s                        | 5 s                  |  |
| flow, 1000 hPa)             | Cold sensor < 10 s                           |                      |  |

Table 3 Humidity sensor manufacturer specifications (based on Table 2 from Vaisala, 2014 and Windsond Catalogue,2016)

| Sonde Characteristics       | RS41-SG radiosondes                 | S1H2 Windsond        |
|-----------------------------|-------------------------------------|----------------------|
| Pressure                    |                                     |                      |
| Sensor type                 | GPS-derived                         | MEMS pressure sensor |
| Range                       | Surface to 3hPa                     | 1100 - 300 hPa       |
| Accuracy                    | Defined as combined uncertainty and | 1.0 hPa              |
|                             | reproducibility                     |                      |
| Resolution                  | 0.01 hPa                            | 0.02 hPa             |
| Combined uncertainty in     | 1.0>100 hPa                         | Not Available (to be |
| sounding                    | 0.3<100 hPa                         | assessed)            |
|                             | 0.04<10 hPa                         |                      |
| Reproducibility in sounding | 0.5>100 hPa                         | Not Available (to be |
|                             | 0.2